# A Tiny Viral Protein, SARS-CoV-2-ORF7b: Functional Molecular Mechanisms

**DOI:** 10.3390/biom14050541

**Published:** 2024-04-30

**Authors:** Gelsomina Mansueto, Giovanna Fusco, Giovanni Colonna

**Affiliations:** 1Dipartimento di Scienze Mediche e Chirurgiche Avanzate, Università della Campania, L. Vanvitelli, 80138 Naples, Italy; gelsomina.mansueto@unicampania.it; 2Istituto Zooprofilattico Sperimentale del Mezzogiorno, 80055 Portici, Italy; giovanna.fusco@izsmportici.it; 3Medical Informatics AOU, Università della Campania, L. Vanvitelli, 80138 Naples, Italy

**Keywords:** SARS-CoV-2-ORF7b, COVID-19, interactomics, topological analysis, cluster analysis, co-regulation network, transcription factors, microRNA, SARS-CoV-2 inter-tissue diffusion, programmed death, SARS-CoV-2

## Abstract

This study presents the interaction with the human host metabolism of SARS-CoV-2 ORF7b protein (43 aa), using a protein–protein interaction network analysis. After pruning, we selected from BioGRID the 51 most significant proteins among 2753 proven interactions and 1708 interactors specific to ORF7b. We used these proteins as functional seeds, and we obtained a significant network of 551 nodes via STRING. We performed topological analysis and calculated topological distributions by Cytoscape. By following a hub-and-spoke network architectural model, we were able to identify seven proteins that ranked high as hubs and an additional seven as bottlenecks. Through this interaction model, we identified significant GO-processes (5057 terms in 15 categories) induced in human metabolism by ORF7b. We discovered high statistical significance processes of dysregulated molecular cell mechanisms caused by acting ORF7b. We detected disease-related human proteins and their involvement in metabolic roles, how they relate in a distorted way to signaling and/or functional systems, in particular intra- and inter-cellular signaling systems, and the molecular mechanisms that supervise programmed cell death, with mechanisms similar to that of cancer metastasis diffusion. A cluster analysis showed 10 compact and significant functional clusters, where two of them overlap in a Giant Connected Component core of 206 total nodes. These two clusters contain most of the high-rank nodes. ORF7b acts through these two clusters, inducing most of the metabolic dysregulation. We conducted a co-regulation and transcriptional analysis by hub and bottleneck proteins. This analysis allowed us to define the transcription factors and miRNAs that control the high-ranking proteins and the dysregulated processes within the limits of the poor knowledge that these sectors still impose.

## 1. Introduction

This study aims to show the effects of the ORF7b viral protein of SARS-CoV-2 on humans, using significant experimental virus–host molecular interactions from BioGRID. Studying protein–protein interactions that contain information and metabolic strategies used by both the virus and its host allows us to understand functional relationships. We performed the analysis after functional enrichment to amplify less represented biological functions. SARS-CoV-2 encodes its genetic information in a single-stranded RNA, and ribosomes translate it into thirty-one different proteins. Viral action occurs through interactions with single human proteins or with protein complexes. To implement an effective action, the total number of viral proteins must be adequate for that of humans. About 5000 viral particles are present in a single human cell during the peak time of infection (the first 3–4 days), along with a concentration of about 150,000 proteins/cell necessary for effective action, as estimated by reliable sources [1,2]. Other estimates [3,4] suggest that in the human cell, there are on average between two and four billion proteins, represented by a few thousand different types [3,4,5], and the average lifespan of each molecule is often measurable in a few dozen minutes. All this implies that each viral protein should interact with a target that has a rather limited time window, but viral proteins also have the same problem because of their turnover rate. Therefore, only a perfect knowledge of human metabolism, deriving from a co-evolution of coronaviruses with humans and/or mammals, can generate proteins that are effective. ORF7b is one of the smallest proteins of the virus [6], an accessory protein of only 43 amino acids with a central alpha-helical segment, but its function is still unknown [6,7]. In recent years, various laboratories dedicated their activities to the research, purification, and characterization of the physical complexes between ORF7b2 and human proteome proteins with different methods and technologies. BioGRID [8] has collected and cured these experimental results within the “BioGRID COVID-19 Coronavirus Curation Project”. BioGRID curates proven protein interactions between viruses and humans, and curators have classified the proteins according to criteria of statistical reliability. They have identified 2753 physical interactions and 1708 interactors for ORF7b (accessed in July 2023). Thus, BioGRID presents an interactome of considerable interpretative complexity for this protein [9].

## 2. Materials and Methods

### 2.1. BioGRID

It is the source of experimental interactions of SARS-CoV-2-ORF7b (Version 4.4.223 as of July 2023). (https://thebiogrid.org/4383871/summary/severe-acute-respiratory-syndrome-coronavirus-2/orf7b.html (accessed on July 2023)). It also collects all the experimentally proven data on the interactions between the 31 SARS-CoV-2 proteins and the human proteome. The quantitative SAINT analysis was used to identify SARS-CoV-2 viral–host proximity interactions in human or model system cells [10,11,12,13,14,15,16], and those with a Bayesian FDR ≤ 0.01 were high confidence. Scores are the sum of peptide counts from four mass spec runs with a higher score indicating a higher degree of connectivity between proteins.

### 2.2. STRING

STRING [10,17] (https://string-db.org/ (accessed on July 2023)) is a database of known and predicted PPIs. The curated interactions are direct (physical) and indirect (functional) associations. The interactions came from different sources (genomic context, high-throughput experiments, co-expression, previous knowledge, etc.), which are channeled into seven independent channels. In this paper, we established the PPI network according to Version: 11.5 of the STRING database. We constructed PPI networks by mapping proteins to the STRING database with a confidence score of >0.9 (highest confidence) by informing seven sources.

Protein enrichment is to some extent based on prior knowledge, and the statistical enrichment of the annotated features may not be an intrinsic property of the input. We used a selected set of proteins by BioGRID as functional seeds. Using Cytoscape software, (Version 3.10.1) we visualized and analyzed PPI networks, which offer diverse plugins for multiple analyses. Cytoscape represents PPI networks as graphs with nodes illustrating proteins and edges depicting associated interactions.

### 2.3. CYTOSCAPE and Network Topology Analysis

Cytoscape [11,12] through Network Analyzer was used to analyze the topological parameters of networks. We examined network architecture for topological parameters such as the clustering coefficient, centralization, density, network diameter, and so on. Our analysis included undirected edges for every network. We termed the number of connected neighbors of a node in a network as the degree of a node. P(k) is used to describe distributing node degrees, which counts the number of nodes with degree k where k = 0, 1, 2, … We calculated the power law of the distribution of node degrees, which is one of the most crucial network topological characteristics. The coefficient R-Squared value (R^2^), also known as the coefficient of determination, gives the proportion of variability in the dataset. We also examined other network parameters, including the distribution of various topological features. We calculated hub and bottleneck nodes based on relevant topological parameters. By examining the PPI network, we found the top 7 hub nodes. These nodes had higher degree values than the others and were in two central modules that were connected and compact.

CentiScaPe is used to calculate centralities for undirected, directed, and weighted networks. CentiScaPe [13] computes specific centrality parameters describing network topology. These parameters assist users in locating the most important nodes within a complex network. The computation of the plugin produces both numerical and graphical results, facilitating the identification of key nodes even in extensive networks. Integrating network topological quantification with other numerical node attributes can cause relevant node identification and functional classification, as well as the topological location of proteins in their specific cellular compartments.

### 2.4. Evaluation of the HUB-and-Spoke Model

Many properties of a scale-free network depend on the value of the degree exponent of the power law γ [14]. Therefore, it is interesting to establish how network properties vary with γ. Estimating the expected maximum degree (also known as the natural cut-off) for a scale-free network, which represents the expected size of the largest hub, is based on the following formula [15]:K_max_ ~ K_min_ 𝒩 ^1/γ−1^(1)
where K_max_ and K_min_ are the expected maximum and minimum degrees of a node. 𝒩 is the system size in terms of the number of nodes.

### 2.5. Cluster Analysis

For the cluster analysis, we used the K-Means Clustering method [16]. K-Means Clustering is an Unsupervised Learning algorithm (centroid-based clustering algorithm) used by STRING to group the protein dataset into different functional clusters. Centroid-based algorithms are efficient, effective, simple, and sensitive to initial conditions and outliers. This makes it useful in handling networks. Here, for K, which defines the number of pre-defined clusters, we used the value of 10 after various manual attempts to search the most reliable clusters in terms of compactness, metabolic functionality, and *p*-value.

### 2.6. GO and KEGG Pathway Analyses

To better research and show the biological function of proteins, we performed GO analysis, which included biological process (BP), cellular component (CC), and molecular function (MF). When the *p*-value was below 0.05, we considered the results to have statistical significance.

### 2.7. Network Analyst—Comprehensive Gene Expression Profiling via Network Visual Analytics: TFs and miRNAs

Network Analyst [18,19] interprets gene lists in a network. It enables the analysis of results present in the network via a powerful online network visualization framework. In protein–protein network analyses, the system also involves the existing relationships between genes, proteins, miRNAs, and human transcription factors, creating a co-regulatory network that is very useful for understanding the mutual relationships between these biological actors.

Information and data used come from various sources.

For gene-miRNA interactions, we used miRTarBase v8.0. It is a database that provides experimentally validated information on microRNA–gene interactions. It contains over 670,000 unique interactions.

For gene-TF interactions, data for target genes were derived from ENCODE ChIP-seq-Data. The BETA Minus algorithm was used to select only peak intensity signals < 500 and, for interactions, predict potential regulatory scores < 1 from ENCODE ChIP-seq-Data.

For signaling, the public repository SIGNOR 2.0, where the signaling data are based on data from the SIGnaling Network Open Resource.

RegNetwork—Regulatory Network Repository of Transcription Factor and microRNA Mediated Gene Regulations. RegNetwork is a data repository of the following five-type transcriptional and posttranscriptional regulatory relationships for humans and mice:tF → TF.TF → gene.tF → miRNA.miRNA → TF.miRNA → gene.

This repository integrates curated regulations and the potential regulations inferred based on the transcription factor binding sites. The transcription factor (TF) and microRNA (miRNA) function at the transcriptional and posttranscriptional levels. It is valuable for studying gene regulatory systems by integrating prior knowledge of the transcriptional regulations between TF and target genes and the post-transcriptional regulations between miRNA and targets. We can also use conservation knowledge of the transcription factor binding site (TFBS) to link the potential regulatory relationships between regulators and their targets. Therefore, from RegNetwork, we can query and identify the combinatorial and synergic regulatory relationships among TFs, miRNAs, and genes [20].

### 2.8. Protein Intrinsic Disorder and Secondary Structure Prediction

We used two online servers, Jpred 4 and IUPred2A. Jpred is a web server that takes protein sequences and from these, predicts the location of secondary structures using a neural network called Jnet. It shows the prediction as a graph. IUPred2A [21,22] is a combined web interface that allows for the identification of disordered protein regions using IUPred2 and disordered binding regions using ANCHOR2. IUPred2A can identify disordered protein regions by analyzing their sequence, regardless of whether they are stable. Upon inspecting the graphic outputs of both predictive systems, we have confirmed disordered segments in most of the examined proteins, whether viral or human, as shown in Appendix A.

### 2.9. SARS2-HUMAN Proteome Interaction Database (SHPID)

In a single database, we collected all the files made available online by BioGRID, containing all the curated physical interactions of the 31 SARS-CoV-2 proteins gained through experiments in human cellular systems with viral baits, followed by purification and characterization with mass spectrometry. These data are available as multiple zip files comprising interactions and post-translational modifications for each single SARS-CoV-2 protein for a total of 33,823 interactions (as of June 2023). The database therefore contains the set of all real interactions existing between the SARS-CoV-2 proteome and all the proteins of the human proteome. BioGRID curators have selected only physical interactions that show a significant statistic with an F.D.R. ≤ 0.01. This means that the probability of a false positive is always less than 1%. Therefore, our database contains all biologically significant interactions known today between SARS-CoV-2 and its human host. The database also contains interactions between individual viral proteins, where known. As part of database search actions, we can ask who interacts with whom through queries using single human or viral proteins. The search can include multiple sets of proteins.

### 2.10. Highlighting the Nodes of a STRING Network Involved in the Same Biological Process (GO)

STRING makes all the nodes involved in the same biological process visible, as evidenced through its mapped databases onto proteins (GO, KEGG, REACTOME, and so on) by activating the process itself with a click of the cursor on the process line. When activated, nodes within the same metabolic process show similar staining patterns. When nodes are involved in multiple processes, they are colored multiple times. This tool is very useful when one wants to analyze multiple nodes involved in many metabolic processes by distinguishing the effect of different processes among nodes and identifying which nodes represent the crossing points. If individual nodes do not show any coloration under the effect of clicking, this identifies certain components of a path, or group, that a specific activated process does not influence. The relationships that determine the coloring of the nodes depend on the knowledge base that STRING organizes for a specific network by extracting data and information from the scientific literature in PubMed.

### 2.11. Comparison of GO Functions in Enriched Networks

From network analysis, STRING defines the enriched biological terms using two numerical parameters. Strength is a measure of how large an enrichment is and is expressed as Log10 [Log10 (observed/expected)], and the False Discovery Rate (fdr) is a measure of the statistical significance of an enrichment. The higher the strength value, the greater the biological effect due to the genetic enrichment with an increased gene expression, while the smaller the *p*-value, the greater the certainty that the event will occur. Often, these characterizations show very different numerical values of strength and fdr. For example, a very low value of fdr tells us that the event is very probable. But, if it is also combined with a low strength value, it means that, despite the high probability, there is not sufficient gene expression to implement the biological event. To decide which GO functions, among the most probable ones, were also those with the best genetic activity, we compared them one by one. Then, we introduced the product P, where,
P = −1[(strength value) × (log10 *p*-value)]
and is calculated from the STRING values to get a quick and accurate quantitative comparison. For example, two GO functions, one characterized by S = 0.35 and fdr = 1.0 × 10^−11^, and another characterized by S = 1.9 and fdr = 1.0 × 10^−6^, could lead one to think that the first is more significant. If we calculate P, we obtain 3.85 and 11.4. This tells us that the increase in gene expression in the second case is functionally prevalent. As reported by STRING, strength = 1 means a 10-fold genetic enrichment with an equivalent increase in gene expression. Furthermore, all fdr values reported by STRING for its biological characterizations (GO, KEGG, etc.) are always statistically consistent with values ≤ 5 × 10^−2^.

## 3. Results

### 3.1. Source of the Data

Fundamental experimental data supporting the role of SARS-CoV-2 in human infection continue to accumulate. BioGRID, one of the most important biomedical interaction repositories, compiled comprehensive datasets of all physical interactions between the proteins of SARS-CoV-2 and the human proteome through the BioGRID COVID-19 Coronavirus Curation Project [8,23]. Curators chose interaction data derived from purification processes in which researchers employed physical methods like Affinity Capture–MS and Proximity Label–MS. Interactions and molecular interactors were classified into various levels of significance. With protein ORF7b (P0DTD8-NS7B_SARS2, UniProt), BioGRID classified 1708 unique curated physical interactors [24,25,26,27,28,29,30] involved in 2753 interactions (accessed in July 2023). Their distinctiveness lies in their ability to avoid repetition and engage in high-confidence interactions at an impressive pace, resulting in remarkable scores in statistical filtering. SAINT (Significance Analysis of INTeractome) express version 3.6.0 confirmed this [24,25,26,27,28,29,30].

### 3.2. Representing ORF7b Data Using Interactomes

Figure 1 shows the circular network of human ORF7b-interacting proteins calculated by BioGRID. Since not all physical interactions flow into a real biological function, the concentric representation of the nodes shows different levels of reliability. Therefore, we used the densest layers as functional seeds. The nodes selected in this study have proven physical interaction through at least two different physical methods. The interaction was chosen to be non-redundant and high-throughput with optimal statistical significance between BioGRID levels 6 and 4. These options allowed us to select nodes with curated unique interactions.

In Appendix A, we show an ARBOR representation of the network calculated by BioGRID with a minimum evidence value of 4, which illustrates the level/association relationships very well. An interactome shows the one-to-one mapping of all interactions, which turns the interactome into an information system [31]. The goal is to decode the functional information of this biological map, the macroscopic properties of which are unpredictable and emergent properties of the system [32,33]. Its inherent complexity makes it difficult, if not impossible, to decode individual hidden molecular information. The datasets curated by BioGRID for each SARS-CoV-2 protein represent suitable starting material. The list of 75 ORF7b interactors with significant levels ranging from 6 to 4 is available in Appendix A. Through the STRING platform [34], we calculated the corresponding interactome (Appendix A) with a score of 0.9 and with all seven data source channels active, to gain as much information as possible. But the graph showed 54 unconnected proteins (72%). So, we added 500 first-order proteins to enrich the interactome and increase the functional relationships (Appendix A). In this new graph, we also eliminated some parental proteins that were still disconnected, leaving 51 final parental proteins that were the basis of our enriched interactome. Network pruning helps eliminate artifacts caused by noisy information [35], while enrichment helps amplify those biological processes that are difficult to define because of their poor representation. Figure 2 shows the interactome obtained after pruning and enrichment. The interactome now appears compact, with all nodes connected. Proteins that share similar functional information should appear as compact sets of nodes and edges (sub-graphs) that perform one or more macroscopic functions. Sub-graphs contain molecular partners that have relational links and perform similar functional activities. Analyzing metabolic processes with Gene Ontology or KEGG allowed us to evaluate the increase in functional annotations.

Many rather compact peripheral modules with a large and very compact central module characterize this interactome. The peripheral modules suggest functional protein complexes. For example, the module at the top of the figure contains a high abundance of ribosomal subunits and is close to many proteins belonging to the translocon complex. The complex on the right is rich in ATPase subunits characteristic of the proton-transporting vacuolar protein pump (V)-ATPase, required for the acidification of secretory vesicles. These complexes represent the set of metabolic machinery necessary for normal cellular life. Surprisingly, the large central component shows intra-connected nodes representing a significant fraction (37%) of the network’s nodes. Components with these characteristics are called Giant Connected Components (GCCs) [36]. This type of component is often present in scale-free networks, in which it is an important substructure. GCCs control the topological growth of the network and thus its evolution [37]. Its capacity to aggregate new nodes and functions makes it a very compact system with a notable increase in the interaction turnover rate of new proteins [37].

We demonstrate this compactness in Appendix A. The figure shows the distribution graph of the mean shortest paths as a function of the degree of the single nodes. The 30 nodes with the highest ranks, i.e., with the greatest connectivity in the network, are those with the lowest average shortest path length. These nodes are all concentrated in the GCC. Thus, this network has a “giant component”, where almost every node is reachable from almost every other node in the GCC through a dense net of interactions. New nodes will join the GCC in a non-linear and unpredictable way to create biological functions, as the GCC is a set of very functionally attractive metabolic nodes. This helps create the set of functions of this metabolic module [37]. As the network grows, the giant component will continue to incorporate a significant fraction of incoming nodes. This means that we should find the main and crucial functional activities integrated into this subgraph.
biomolecules-14-00541-t001_Table 1Table 1Topological parameters calculated for the interactome of Figure 2.Summary Statistics of the Network *NotesNumber of nodes551
Number of edges4648**Avg. number of neighbors16,871Average connectivity of the nodesNetwork diameter9
Characteristic path length3.666
Clustering coefficient0.5490 ≤ C ≤ 1Network density0.031
Network heterogeneity1.057Tendency to contain hub nodesNetwork centralization0.259The extent to which certain nodes are far more central than othersConnected component1***(*) Calculated by Cytoscape Network Analyzer, which computes a comprehensive set of topological parameters [38,39]. (**) Most nodes (77%) with a score of 0.9 contain a very large component of the scientific information necessary to calculate the interactions that derive from the Text Mining channel with only a partial presence of data coming from the Experiments channel. However, only 15.7% of the edges show a full score of 0.9, deriving from the “Experiments” channel alone, proving that their interactions are experimental. (***) This value is “1” to show that all nodes in the network are connected to each other. Existing unconnected components (CC > 1) alter the calculations of the topological parameters, making them unreliable. This is the fundamental reason for pruning. A single component accounts for strong network connectivity. Calculation by Cytoscape.

### 3.3. Principal Characteristics of the Interactome

We transferred the interactome to Cytoscape [38] and analyzed it with the help of CentiScaPe (v2.2), Analyze Network [39], and STRING-app [11,40], which generated a table of nodes containing various columns with the quantitative values of many topological and functional parameters. This allowed for the evaluation of characteristic topological and functional features for each node of the interactome.

By examining the value of parameters in Table 1, we can deduce that we are considering a network composed of many independent and compact peripheral modules that interact with few connections between them, although these connections are crucial. The large diameter, network heterogeneity, and low density are all topological parameters that suggest low connectivity and functional independence [41], which is also supported by the discontinuities between the sub-graphs and the central core. In particular, a large diameter suggests a wide system with peripheral components quite distant from the central module. The shortest average path length, which gives the mean distance between two connected nodes in the modules, should be a metabolic advantage because small average lengths minimize transition rates between metabolic states in response to external stress. The clustering coefficient also supports this topology. It is a basic index for local density in a network and is a measure of the degree to which nodes in a graph group together. It takes values of 0 ≤ C ≤ 1; thus, a value of 0.549 shows a tendency to form clusters, where each node shows an average of 16.817 neighbors. This coefficient of aggregation, according to Barabasi [42], decreases with an increase in nodes.

The function used for the fit is f(x) = a x^b^, where the values of a, b, and R^2^ are 0.29, −1.89, and 0.62. The significant *p*-value of 1.0 × 10^−16^ from the interactome analysis and the good correlation index underscore a strong expectation of preferential relations or associations among nodes following their enrichment.

Figure 3 shows the characteristic power distribution of nodes of a scale-free network [37], where the vast majority of nodes have very few connections and only a few (HUBs) have a very large number of connections. This distribution is a distinctive feature of biological networks, regardless of the experimental approach [43], and is important for understanding the behavior of the system. In these networks, the number of nodes necessary to control the entire system is reduced to a minimum, improving functional speed and providing better control of the system [44,45]. Our interactome has an absolute gamma value of 1.89, which favors the formation of large topological modules (GCCs). A topological module represents an area of the network that is densely packed with nodes and links, where nodes have a greater tendency to connect to nodes in the same area than to nodes located outside the area itself. In this scenario, high-degree nodes are the most attractive. Their acquisition rate is faster than the growth of the network in terms of the number of nodes it contains. Here, the dynamic is “winner-takes-all”. This feature is often based on a topological organization of the main nodes of the hub–spoke-type, where all these nodes are located within a short distance to each other [46,47].

In the figure, we highlight the seven HUB nodes (EGFR, SRC, PIK3R1, PIK3CA, GRB2, and HRAS) that have superior ranks compared with all the others, also remembering that the GCC includes the top 30 nodes with the highest ranks. Hub nodes model the architecture of metabolic modules. EGFR, which serves multiple critical functional roles in the cell, is the highest degree interactomic hub node because of its exceptional capacity for PTMs (see Appendix A).

We need alternative information to prove the accuracy of our observations and hypotheses and to decode the information considering the actual functional activities in which ORF7b2 is involved. The following tables show the most significant information obtained from GO analyses. To assess the importance of each functional property, we use the *p*-value as the evaluation criterion [5] for the main significant processes. STRING calculated the tables with the methods and techniques of GO analysis.

### 3.4. Quantitative Evaluation of the Biological Functionalities in the Interactome

Table 2 shows the overall picture of the many functional activities performed by the entire network. Over 10,000 significant PubMed publications were used to provide coherent information on the 5057 functional terms. STRING calculated the entire interactome using this knowledge base. This assures us that the functional relationships taken into consideration are very robust and that the pruning operation reflected real knowledge gaps in the considered node properties. The spectrum of biological activities induced by ORF7b2 appears broad in 15 categories and, therefore, it is difficult to define and study. We evaluated and selected the functional activities from time to time, as each of the 5057 terms reported in Table 2 has a statistical value (*p*-value) that is always less than 0.05, ensuring their significance. In this study, we tried to provide a comprehensive view of the metabolic and molecular activity induced by ORF7b. Future studies will try to go into more detail.

Table 3 shows the most significant biological functions (GO biological processes) among the 1690 related to the human proteome following the action of ORF7b. The principal activities involve the control of intracellular transport, also by vesicles, and the control of their localization in the cell. The set of cellular processes includes the transportation, binding, and holding of a protein complex or organelle in a specific position. A transporter or group of transporters facilitates the directed movement of molecules or cellular complexes into or out of a cell, or between cells, to effect transmembrane, microtubule-based, or vesicle-mediated transport. A significant value ranging from a *p*-value of 1.0 × 10^−77^ down to 0.05 marks all 1690 activities. Enzymes and signaling pathway receptors also appear to be possible prime targets, considering the large number of human proteins involved. In particular, the series of molecular signals started with an extracellular ligand binding to a receptor with tyrosine kinase activity on the surface of the target cell and ended with regulating a downstream cellular process. The statistical significance of these biological actions is very high, as is the number of proteins involved. However, the table shows a comprehensive picture of 1650 functional activities that belong to both the virus and the cell in performing their respective strategies of attack or defense. A part of these activities also refers to the basal metabolic activities required to maintain normal vital functions (housekeeping functions). As we will see later, it is possible to extract the specific activities of the virus.

Table 4 depicts the location in the cell where the most statistically significant functional activities (as presented in Table 3) occur. Many cell membranes, the cytoplasm, as well as protein complexes, are metabolically involved. Of particular interest is the significant activity performed by the SNARE complex, which is involved in driving vesicles and endosomes toward the correct cellular target and also provides the correct docking. SNARE proteins (SNAp REceptor, i.e., Soluble *N*-ethylmaleimide-Sensitive Factor Attachment Proteins) are a family of cytosolic proteins involved in vesicular fusion with the target membrane during intracellular transport and exocytosis [48]. SNAPs interact with proteins of the SNARE complex during the recycling of the fusion complex components [49]. We know that interference with the function of SNAP proteins is associated with many pathological processes, such as colorectal cancer [50], epilepsy [51], and Huntington’s disease [52]. However, it is the post-translational process by which a PTM protein (a proteoform) trans-locates from the ER to its final destination, which drives function. This process also includes tethering and docking steps that prepare vesicles for fusion.

Table 5 (Reactome) shows the most statistically significant molecular mechanisms in which ORF7b might involve the human proteome. It contains biomolecules that perform precise metabolic and signaling activities and their relationships, which are organized into biological pathways. Beyond the various interferences on important metabolic pathways, it is interesting to note the metabolic functions shown, such as nervous system development, immune system, infectious disease, hemostasis, innate immune system, platelet activation, insulin receptor signaling, viral mRNA translation, and cell–cell communications. Although these vital metabolic functions have high statistical significance, it is of great significance that the parallelism with the known clinical effects of COVID-19 on the human organism [53,54] is not overlooked. In fact, they represent significant areas of virus interaction.

The spectrum of possible viral interference might also involve intracellular transport mechanisms and cell–cell communications. Many of these “actions” have a deep impact on human biology and inter-organ signaling, according to recent research on the effects of COVID-19 on the human organism [55,56]. In particular, we relate the most significant one to signaling by receptor tyrosine kinases (RTKs), a family of proteins that act as cell surface receptors for various factors, such as cytokines and hormones. These receptors control many cellular processes and also have a crucial role in the development and progression of many types of cancer [57,58]. It is also interesting to highlight the high significance of this interactome in some activities, such as “cell surface interactions on the vascular wall”, “platelet activation”, “insulin receptor recycling”, “viral mRNA translation”, and “cell–cell communication”.

By using proteins involved with ORF7b, we extracted relevant activities in this interactome from the human proteome. The symptoms of patients with COVID-19, including thrombophilic alterations [59], hyperglycemia [60], and systemic spread of infected cells [61], may not be independent, as their underlying mechanisms, as found in Reactome, all appear to involve ORF7b, which may be the underlying cause.

The number of human tissues and organs that are potential targets of ORF7b is also staggering. Table 6 shows these tissues/organs, which are important constituents of the human body through many cell types.

These tissues/organs share many of the described metabolic activities to varying degrees. Therefore, even if not all, they are potential targets of the virus where it finds optimal metabolic conditions for its replication [62]. The need to expand the list of terms in this table arises from the need to show the many target tissues of the virus with significant potential. It is amazing how a tiny protein like ORF7b could induce such a wide effect. This also means that the protein appears to be an authoritative candidate for altering the molecular mechanisms that keep cells in contact with each other [63,64,65]. Dysregulating these mechanisms might free the cells to spread without a programmed death [66,67].

This table shows a long list of the various organs in the abdominal cavity that are potential targets of the action of this protein and validates the clinical observations that COVID-19 is a systemic disease. The high statistical values suggest the enormous potential of the strategy implemented by SARS-CoV-2 in attacking the human body. Some objectives are of particular interest. The nervous system (central and peripheral), human reproductive system (male and female), placenta and fetus, blood, and hematopoietic system should alert us to the consequences encountered in long COVID. Long COVID is associated with symptoms that suggest including these specific organs and tissues as well.

Another significant index to consider is the large number of proteins taking part in various functional activities, as stated in the tables discussed earlier. Considering the finite number of proteins in the interactome and the large number of them involved in many different metabolic activities, this suggests that there is a high probability that single proteins may be involved in numerous functions. But all this also suggests that, with viral infection, a single human protein can perform many functional activities, some for the benefit of the cell and others for the benefit of the virus. KEGG pathways can infer higher-level functions and metabolic utilities of the human system from genomic and proteomic data. It groups genes and/or proteins into “pathways” as lists of genes/proteins taking part in the same metabolic process. Thus, KEGG is very useful for computational analyses, including metabolic modeling and simulation according to systems biology, and translational research in disease development. The KEGG results show a wide range of activity. More space would be required to highlight the breadth and diversity of many of the responses (195 pathways) and their statistical significance. However, we included the most probable in Table 7. These pathways reflect precise connections with the functions reported in the previous tables, which identify and endorse their metabolic roles. We identified the most significantly represented functions, but we could not at this stage establish a direct correlation to viral activity.

So far, we examined the spectrum of functional/molecular activities present in an infected cell and, in particular, those involved by ORF7b. Once we define the principal functions, we will highlight which single proteins favor the virus by “playing a double game”.

### 3.5. Exploring the Physical Basis of Cytoskeletal Alterations Caused by ORF7b

The propagation of a virus to uninfected cells makes up a crucial phase in its life cycle, achieved by liberating novel viral particles from the infected cell. The ability of ORF7b to induce changes in the cytoskeleton that could promote the spread of infected cells is not coincidental. As we previously discussed, these changes seem to derive from dysregulations induced at the cytoskeleton level. These results, however, suggest different biological events from those already known, not only the spread of viral particles after cell rupture but also the spread of all the infected cells to distant tissues, like that observed in tumor metastases. Therefore, this aspect needs greater attention. The key processes for modifications of the cell membrane, or that of cellular compartments, should pass through direct deformations caused by specific proteins that interact with the membrane [68] or even through indirect deformation by the cytoskeletal structures [69]. Therefore, the cytoskeleton is one of the key driving forces, having a close association with these events [70].

Unfortunately, understanding the influence of these molecular processes on the physical structure of the membrane is still an unsolved challenge, despite a slight improvement in our understanding of the underlying physical basis. Until now, it has been difficult to quantify the forces present in living cells within these processes. However, we now have a first, albeit crude, quantitative understanding of force production and distribution at the molecular level using clathrin-mediated endocytosis as a model [68,69]. During endocytosis, the actin cytoskeleton generates forces that are transmitted to the plasma membrane through a multi-protein coat, leading to membrane deformation. Although the exact extent of these forces remains uncertain, we can highlight a phenomenon of accumulation and redistribution of force within the endocytic mechanism. This has led to the widespread belief that the EPNs and Hip1R proteins transmit the force generated by the assembly of the actin to the plasma membrane [71,72]. As both protein types also attach to clathrin and other coat proteins, it is plausible that transmitting forces to the membrane might occur through multiple pathways [73,74].

However, we know which eukaryotic genes/proteins engage in these processes, serving as either components or regulators of the cytoskeleton, while an intricate interplay between lipids and proteins controls membrane remodeling during intracellular trafficking [75]. Noteworthy examples include MTOR, CTNNA1 (alpha 1 catenin), CTTN (cortactin), ITGBs (integrins), CDH1, CDH2 (cadherins), ACTB (actin B), and EPNs (Epsin family). A review of the interactome in Figure 2 identifies all eight proteins and various members of their families (please also refer to the accompanying Excel file for the comprehensive list and node degrees). This observation drew our attention to the intriguing possibility regarding the potential involvement of specific human proteins, in particular, those associated with cytoskeletal modifications and negative regulation processes, in the mechanism of SARS-CoV-2 spread to non-infected cells and tissues. We used these proteins as seeds to tease out their functional relationships within the human proteome. Figure 4 illustrates the specific and close relationships among them during their involvement in the processes that impact the organization of the cytoskeleton. We used a specific feature of STRING to highlight and color the proteins involved in the same biological process (see Material and Methods, Section 2.10).

The network comprises all the human proteins involved in cytoskeleton dynamics. Since they are all reported in BioGRID as interacting, this suggests direct physical and/or functional associations. Within the high-ranking group, proteins such as ACTB contribute to a single dysregulated process (one color), while proteins like MTOR manage multiple dysregulated processes (various colors). However, these interactions imply that SARS-CoV-2 exploits the host cell’s proteins involved in processes regulated by CDH1, CDH2, EPN1, EPN2, CTNNA1, ITGB1, MTOR, ACTB, CTNNA1, and CTTN. This affects cellular functions related to cell adhesion, signaling pathways, cytoskeletal organization, and programmed death through the Viral Hijacking of Cellular Machinery. But these specific interactions also suggest potential roles for these cellular proteins in stages of the viral life cycle. In fact, their presence shows that these host proteins contribute to SARS-CoV-2 infection dynamics and pathogenesis, thus becoming appropriate therapeutic targets. However, further observations are important. Structural models of protein interfaces and the potential impact of post-translational modifications are crucial to understanding molecular mechanisms based on interactions because alteration of these characteristics might change protein–protein interactions and related biological functions. Many of the cytoskeletal proteins possess disordered structural domains and many phosphorylation sites. MTOR, serine/threonine protein kinase, in the presence of RPTOR (Regulatory-associated protein of mTOR) and RICTOR (RICTR, Rapamycin-insensitive companion of mTOR), and through mTORC1 and 2 complexes, controls the phosphorylation of at least 800 proteins, and the actin cytoskeleton is MTOR-sensitive [76,77]. DEPTOR (DEP Domain Containing MTOR Interacting Protein) is a negative regulator of TOR signaling and the mTORC1 and 2 pathways, inhibiting the activity of both complexes [78,79]. This leads to negative regulation of cell size and negative regulation of protein kinase activity. MTOR, DEPTOR, RICTOR, and RPTOR are all part of the interactome and communicate extensively. Thus, the relationships among them validate the various dysregulations in Figure 4 and Table 8. A last consideration is that another viral protein also interacts with the cytoskeleton. It is the N protein, which plays various roles in the life cycle of the coronavirus [80]. We should emphasize that the N protein has a physical interaction with ACTB [81], resulting in cytoskeleton manipulation similar to other viruses (see also Table 5). We mention the N protein because it plays a role in the formation of liquid droplets, which is an overlooked aspect of this virus.

In Figure 4, the Table on the right side shows the nodes with the highest degree. In the table, we also report CDH2, CTNNA1, and EPN1 to show all seeds. The number of colored segments of each protein node shows the dysregulated processes in which it is involved, as shown in Table 8.

We can conclude that the interaction between the SARS-CoV-2 ORF7b protein and host cell proteins, especially those involved in cytoskeletal modifications, plays a role in the virus’s ability to propagate infected cells to target distant tissues. Structural disarrangements or metabolic dysregulations induced at the cytoskeleton level impact a cell’s ability to counteract viral infection, aiding in viral spread or facilitating intracellular transport of viral components, thus contributing to its long-distance diffusion.

### 3.6. Topological Analysis

When a virus infects a cell, viral proteins represent the attackers and seek vulnerabilities in the network. Vulnerabilities introduce uncertainties into the network as a loss of original metabolic performance, even by changing information flows. Examining the network topology allows us to study both vulnerability and functional uncertainty and to seek any architectural or functional changes. Pathways that cross between metabolic pathways or between signaling pathways are among the most vulnerable topologies, while hub-and-spoke topologies have the least uncertainty in destabilization. Therefore, topological data analysis is a powerful biological network analytic method [46]. To extract meaningful information from interactomic data, it is essential to understand the correlation between topological parameters and the mechanisms of biological functions [82]. Centrality metrics measure prizing nodes by attempting to quantify the idea that some nodes are more “important” than others.

We can divide topology scoring metrics into two groups including a local one to evaluate individual nodes and a global one to evaluate the network. Global metrics include betweenness, bottleneck, eccentricity, closeness, radiality, stress, and more. It is a useful methodological approach to increase the efficiency in selecting, characterizing, and classifying crucial proteins as both hub and/or bottleneck proteins. In particular, bottlenecks are key link proteins, almost always not HUBs, but hard-to-discover essential proteins that control and regulate metabolic crossovers. Being intermediate (bottleneck) in regulatory networks indicates functional essentiality, which is often more significant than being a hub for understanding information flow.

Eigenvector centrality measures the transitive influence of nodes. Relationships originating from high-scoring nodes contribute more to a node’s score than connections from low-scoring nodes. A high eigenvector score of a node indicates its connection to multiple nodes with high scores. Figure 5 (top) shows the distribution analysis of the eigenvectors. The graph shows that the eight highest values have a degree value matching that of the selected eight hub nodes, showing that all hub proteins also have the highest eigenvector scores. Stress is an index of node centrality. It represents the number of the shortest paths passing through a node. A high-stress node is a node traversed by a very large number of the shortest paths. In an interactomic network, it shows the relevance of a protein in keeping communicating nodes together. We can consider such a protein as a “bottleneck” protein [83,84,85]. The higher its value in the network, the more relevant the protein is in linking regulatory proteins of different pathways. However, because of the parametric significance of this index, it is sometimes possible that stress shows a molecule that is only involved in many cellular processes but not relevant for maintaining communication among other proteins [86]. Figure 5 (middle) shows the stress distribution analysis where SEC13, EGFR, MTOR, HSPA5, VAMP2, and SRC are the major stress proteins. Betweenness [82] is also an index of node centrality, similar to stress, but with more information. It is a measure to rank the relative importance of vertices or edges. It represents the total number of non-redundant shortest paths connecting a pair of nodes, i.e., a1 and a2, crossing the node a. The betweenness value of a node increases if it lies on a non-redundant shortest path between nodes a1 and a2. Therefore, a high betweenness score characterizes a key node in maintaining connections, and this type of node becomes the critical point that controls the communication among other distant nodes in the network. In biological terms, it characterizes the interactivity of a protein in an interactome, showing the protein’s ability to link distant proteins. Thus, betweenness is a measure of how important the node is to the flow of information through a network. This feature of the node in a protein signaling network may also show the potential of the protein to act as a bottleneck. It acts as a junction connecting metabolic pathways that can hold the communicating proteins of different pathways together. The higher the value, the greater the potential of the protein as a bottleneck molecule. The interdependence of a protein shows the ability of this protein to link distant proteins. When reporting modules, intermediate relationships are crucial to maintain functionality and consistency in the reporting mechanisms.

The analysis in Figure 5 (bottom) confirms that EGFR, SEC13, MTOR, and HSPA5 are “bottleneck” proteins, and also shows a new protein, SEC61A1. In the stress distribution, the SEC61A1 value was very close to that of VAMP2, while now the VAMP2 value is close to that of SEC61A1. Therefore, we can consider both proteins as bottlenecks. In a multi-parametric approach, we used eigenvector, stress, and betweenness centrality distributions to validate the eight hub proteins and define the role of some proteins as bottlenecks. Among the proteins selected as the most ranked bottlenecks (EGFR, HSPA5, MTOR, SEC13, SEC61A1, SRC, and VAMP2), EGFR and SRC show a dual role, both as a hub and as a bottleneck. Putting it all together, we have EGFR and SRC, which are mixed (HUB/bottleneck) proteins, HSPA5, MTOR, SEC13, SEC61A1, and VAMP2, which are pure bottleneck proteins, and PIK3R1, PIK3CA, GRB2, and HRAS, which are pure hub proteins. These differences allow these proteins to be defined into three classes of molecular markers. In a eukaryotic protein interaction network, a node represents the lone native protein because of alternative splicing [87] and proteoforms [88]. This may be a problem because, in all databases (including STRING), it is customary to collapse all the different functions of its isoforms and proteoforms onto the native protein, attributing it to a greater load of functions that it does not possess. In the interactome calculation, this anomaly produces biased nodes with higher and unreal connectivity.

Researchers have identified three different types of hubs in tissue-specific protein–protein interaction networks as follows: a few tissue-specific hubs, many tissue-preferred hubs that are formed by connected proteins, and housekeeping hubs that are involved in normal metabolic management [89]. When we connect these features to their specific functional roles within different tissues, they exhibit distinct functional differences, which are influenced by structure/function relationships.

Disordered regions enrich pure hub and hub/bottleneck proteins among the three previous classes, and as a result, these proteins harbor a significant number of predicted binding sites [90]. They are also rich in splice variants, have longer peptide chains, and host a significant number of domains. This successful structural versatility drives their high propensity for interactions [88]. Because they are involved in essential functions such as phosphorylation and mRNA slicing processes, they get tangled in multiple intracellular functional pathways. Pure bottleneck proteins are extracellular proteins that are connected to pathological conditions, such as cancer, and play a role in cell-to-cell signaling pathways. Defining the actual functional role of a node is challenging because of the convergence of multiple functions with varying spatio-temporal characteristics. Many researchers still use static and deterministic approaches to select their experimental design, which leads to these limitations.

The topological role of network hubs depends also on the exponent value of the power law [91]. The value of <2 for the degree exponent b (see Figure 3), although very close to 2, suggests a hub-and-spoke architectural model. The hub–hub network of the entire interactome fits a hub-and-spoke model, as Perera [14] and Barabasi [15,42] suggest. The largest hub (EGFR, 159 nodes) acts as a central coordinator and connects to a significant portion of nodes, which is shown in Figure 3 and Figure 5. These structures act as a backbone connecting different metabolic modules. In this topological context, we should also identify the top hubs as significant centers of control over the entire network. This view also agrees with the topological parameters calculated by the Cytoscape Network Analyzer.

Figure 6 also shows the relationships and the particular topology involving both HUB and bottleneck nodes [92]. Appendix A shows that EGFR organizes in a topologically similar manner, even when conditions are normal. Relationships among the HUB nodes are strong, while those with the bottleneck nodes are less intense, as the figure shows. All these significant nodes play a collective role in maintaining the stability of the hub-spoke system, albeit with varying functions and methods [46]. Each of them controls many different biological processes [47]. The following question remains: which node, regardless of its degree, is involved in the greatest number of functional processes? The question is not far-fetched. Because of the many metabolic crossroads, greater connectivity may not correspond to greater functional involvement [93]. When designing a drug, it is important to have this information.

While surprising considering the very high number of functional involvements, Table 9 shows how a HUB node is not always the main controller of the metabolic landscape. MTOR (degree = 24) and HSPA5 (degree = 19), although with lower connectivity, are involved in a very significant number of processes. The node distribution and biological functions in the hub-and-spoke system, coupled with the ORF7b-induced interactome’s complexity, handle this outcome. The next inquiry is how functionally significant are the processes they regulate. The answer would require a large analysis not covered by this study. Certainly, these same nodes, depending on their level of genomic expression, can both up-regulate and down-regulate a biological process [94,95,96]. Down-regulated processes, or “negative biological processes” according to GO, are important to highlight because of their higher probability of resulting from viral strategy [97]. Here, as we will see below, statistical significance is no longer the only parameter to follow.

### 3.7. The Functional Effects Depend Not Only on ORF7b but Also on the Integrated Action of Several Viral Proteins

The virus shows extraordinary strategic potential. Our previous results showed the specific impact of its proteins on crucial metabolic processes. About 200 patient symptoms [98] were used to generate various hypotheses based on clinical impressions found to be associated with long COVID. All this shows how broad and diversified the systemic action of the virus is. Thus, part of the broad spectrum of metabolic activities found in this interactome might be associated with the multitude of clinically observed symptoms [99]. However, we should not underestimate the vast potential of the ORF7b protein. The proteome yields biological functions via target proteins, which result from specific one-to-one interactions between viral and human proteins. Other viral proteins could target human proteins present in metabolic modules where ORF7b also operates. The ORF7b circular interactome (Figure 1) displays other viral proteins, including ORF3a and M, which may show their ability to target human proteins in the same metabolic modules as ORF7b. As of July 2023, we had organized a database called SHPID, which contains BioGRID interactions. In this database, we collected 33,823 interactions between SARS-CoV-2 and human proteins. We analyzed the hub proteins highlighted in Figure 3. The proteins EGFR, SRC, and PIK3R1 are the major HUB nodes of the ORF7b interactome with 159, 123, and 90 links. Although these proteins are involved in the ORF7b interactome, Table 10 reveals that they also interact with other viral proteins.

Table 10 depicts how these high-degree human proteins are a common target for many viral proteins. Our analysis of the interactions between the thirty-one viral proteins and the human proteome, as reported by BioGRID, yielded this result. Even though viral proteins have co-evolved with their human host or other species, they seldom possess structurally detailed molecular interfaces for accurate and stable interactions. Only a few viral proteins exhibit strong interactions, akin to those observed in complexes. Most of the interactions have weak bonds, mostly because of the anisotropy of the contact areas [101]. Viral proteins attempt to establish competition with normal binding proteins by mimicking interaction interfaces to the greatest extent possible, binding to target proteins with interaction constant values that typify weak processes. The interfaces mimicked by viral proteins compete through multiple and transient cellular interactions. They interact with hubs and bottlenecks in the human PPI network to control vital proteins in complexes and pathways. Proteins can overcome structural difficulty by introducing an intrinsically disordered region (IDR) into the sequence, which can enhance the mimicry of contact surfaces. IDPs have IDR stretches that may be part of low-affinity inter-molecular interactions [102]. With the emergence of IDPs in eukaryotic proteomes [79], the disorder becomes crucial information for PPI evaluation.

Many of the interacting viral proteins in Table 10 show IDRs (see Section 2); thus, the probability of multi-targeting is high, and this could explain the phenomenon (see also Section 2 for details). After all, even the three human proteins analyzed have disordered and mobile segments. They are lipid-anchored proteins with a central body in the cytoplasm or outside the cell. Two long disordered and mobile tails are present in EGFR, which is found on several internal membranes (endosomes, ER, Golgi, nucleus) and the surface. SRC also has long disordered and mobile tails, some mobile central segments, and multiple localizations, both on the surface and on intracellular organelles (endosomes, mitochondria, etc.). PIK3R1 also shows a long-disordered C term with many mobile intermediate segments and is on the cell surface. To this, we should add that the disordered/mobile parts often show PTM sites. The presence of PTM sites increases the number of proteoforms for any single protein, increasing the probability of interacting with new molecular partners and establishing new functions.

A particular observation is that our database shows that ORF7b itself interacts with the viral N protein (see Table 10). Among the various functional peculiarities of this protein, we find that it is involved in forming liquid droplets [80]. The liquid–liquid phase separation is a key mechanism for organizing macromolecules, such as proteins and nucleic acids, into membrane-free organelles [103]. The N protein can self-bind into spherical aggregates, which can diffuse in the condensed phase and exhibit liquid-like behavior [104,105].

Although we also examined other relevant human HUB nodes of the ORF7b interactome, such as PIK3CA, EGF, and HRAS, we did not find other direct targeting of viral proteins. Therefore, these seem to be nodes extracted from the ORF7b functional enrichment that are functionally connected with the other HUBs of this network. Thus, their presence in this interactome seems to be a specific functional requirement of ORF7b. After all, the human metabolic system responds to the ORF7b protein, consistent with the multiple metabolic responses of multicellular eukaryotic systems. In particular cases, viral action may require the synergistic action of different viral proteins. Thus, to achieve its biological effect, the virus can also use complex and sequential interaction modes on a single protein. This analysis is in excellent agreement with the previous classification of hub and bottleneck proteins. Unfortunately, we do not know where, how, or when these interactions occur. Hence, our vision of a dynamic phenomenon is only static and somewhat unclear, which may also be spatio-temporally inappropriate or distorted in our reconstruction of it [106]. Nevertheless, SARS-CoV-2 employs a known strategy of targeting the same human protein with multiple viral proteins [107].

### 3.8. The Peculiar Case of GRB2, a Protein in the Service of ORF7b

GRB2 (Growth Factor Receptor Bound Protein 2—UniProt: P62993) is a protein that, according to BioGRID, binds ORF7b, although with the low level 1. Our observation within the BioGRID dataset reveals that this protein only interacts with ORF7b. It was not included in the seed proteins because of its low significance, but we found it recruited in the interactome [108]. The enrichment suggests that this ORF7b interactor is essential for virus infection. It assumes the role of a HUB with 84 connections and controls 233 biological processes (see Table 11). GRB2 is an important protein that provides a critical link between the phosphorylated cell surface growth factor receptors (EGFR) and the PI3K-Akt signaling pathway. Both the KEGG and Reactome pathways reported its significant involvement in several signaling mechanisms (hsa04151, the PI3K-Akt signaling pathway; HSA-1963640, GRB2 events in ERBB2 signaling; HSA-179812, GRB2 events in EGFR signaling; HSA-354194, GRB2:SOS linkage to MAPK signaling). Later, we came to know that it is often involved in various dysregulation processes that assist viral activity. Table 11’s proteins and GRB2’s case show the sophisticated and diverse molecular strategy of SARS-CoV-2. The hubs listed in this table are proteins obtained through functional enrichment, but they are not direct molecular interactors of ORF7b.

### 3.9. The Role of ORF7b

The diverse and sometimes contrasting metabolic properties of some of the interactome nodes are surprising. Among the 1691 biological processes (GO) induced by ORF7b, there are 117 peculiar metabolic activities mentioned as negative activities (7%). Most of the HUB and bottleneck proteins are also involved. According to AmiGO-2, the official web-based set of tools for searching and browsing the Gene Ontology database, negative activity means “any process that stops, prevents or reduces the frequency, rate, or extent of metabolic functions”. *p*-values alone cannot guide us in identifying which terms are most significant for these purposes. STRING measures the size of the enrichment effect by also using the “strength” score. The sole use of the *p*-value can produce an overrepresentation of the GO term, while the value of P (see the Section 2) is useful for amplifying those underrepresented biological processes connected with a specific context [109] through their expression. The limit of this approach is that, in a complex interactome, many proteins are not specific to a single metabolic pathway but are sometimes even part of multiple pathways. Here, the massive study of some of these pathways favors assigning the protein to the more studied GO pathways. In fact, the databases favor assigning the protein to the more studied GO pathways and obscure the emerging relationships towards different biological pathways that are not studied or represented [110]. Therefore, the analysis should select only the most reliable terms.

In addition, Hong et al. [110] showed that linked gene pairs, even of different functional pathways, according to KEGG, show positive expression levels. Therefore, these two genes (or their proteins), even in a functional pathway altered by disease, are up-regulated or down-regulated together. This is due to their reciprocal and close functional relationships [111]. So, when a disease affects a metabolic pathway, all the genes in the pathway will regulate their expression. Therefore, an over-representation of a GO process suggests an over-expression of the genes and their decoded products that make up the metabolic pathway since they have close functional relationships with each other in regulating the expression [110,111].

We selected 17 terms with the highest possible strength value, paired with a very significant *p*-value, and listed them according to the value of P (see the Materials and Methods, Section 2.11). Table 11 reports these terms according to the expressed rule. In the table, among the proteins involved in these negative functional activities, we note (in bold) many of the proteins highlighted as HUB nodes or “bottlenecks” or as involved in other important signaling pathways. Although all biological processes show positive values of enrichment (high strength), very many have minimal or negligible enrichment. It is necessary to exceed the value of 0.5 to have an enrichment of 3 times. We found that 32.28% of the processes have enrichments lower than 3 times and only 14.7% have enrichments greater than 10 times. The remaining 53.02% has intermediate enrichment values between 3 and 10. This means that there are very few of the most enriched fractions, and we can think of the average enrichment of most biological processes as being suitable for the normal metabolic function to be performed. The 17 selected terms therefore make up a very limited set, less than 1%, but it is the only one that can boast a significant and even conspicuous enrichment. However, the negative term means over-enrichment and, therefore, suggests a gene over-expression. Some sets of proteins, enriching themselves, change their functional state, inducing changes in the pathways they control. Since the term negative means loss of control, down-regulation, or a weakening of their functions, the deactivation (or even activation) of the functional pathways they control is favored. This is not new. We find a dysfunctional expression of genes with over-expression and deleterious functions during disease or even aging, in particular, for genes involved in pathways related to stress responses, antioxidant defenses, and DNA repair [112,113,114].

Our examination of Table 11 enables us to affirm that many pathways show significant dysregulation, suggesting that we may have identified pivotal genes associated with these pathways. At present, describing what occurs is challenging because of the lack of data to pinpoint causes, determine the opportune moment for the process, and establish the sequence of events, which is due to the lack of space–time information. The strategy of ORF7b, in collaboration with other viral proteins, aims to create a viral microenvironment that helps infected cells minimize cell matrix rigidity and adhesion, increase intracellular oxidative stress, generate pro-survival signals, trigger the epithelial–mesenchymal transition process, and inhibit intracellular transport and ER activity, starting widespread cellular metabolic deregulation. We should emphasize that the process of metastasis, characterized by the epithelial–mesenchymal transition (EMT) and its inverse, the mesenchymal–epithelial transition (MET), plays a crucial role in the metastatic spread of carcinomas [115]. Likewise, these events appear to be among the primary targets in preventing the programmed cell death mechanisms of infected cells, allowing survival after separation and systemic spread.

In particular, we can see the dysregulation of all protein tyrosine kinase receptor activities. This reduces the processes of the internalization of external signals and the activities of receptors activated by growth factors. The integrated dysregulation of oxidative stress, the unfolded protein response of the ER, and lysosomal action [116,117] also favored integrin-mediated alterations of the intercellular matrix and the loss of control over cell-extracellular matrix adhesion processes. The intention behind all these activities is to dysregulate programmed death processes such as apoptosis and anoikis, promoting the spread of infected cells in the body [118].

The systemic spread of infected cells explains why the tissues and organs shown as infectible in Table 12 are so numerous and significant. In the presence of infected cellular material widespread in the body, the virus has also the potential ability to cause inflammatory processes in the brain, so it is important to pay particular attention to the dysregulation of blood–brain barrier permeability. By altering endocytosis, endosomal trafficking, lysosomal degradation, blocking anabolic processes, and lipid transport, this creates mitochondrial dysfunction, resulting in a heavy dependence on glucose for energy production. Many miRNAs work within the cell and could interfere with these procedures. However, distinguishing them through this type of analysis is not yet possible.

In a nutshell, this tiny protein is involved in controlling the intercellular communication of the virus. By suppressing intracellular signaling, it created a metabolic microenvironment that caused generalized metabolic dysregulation and blocked the intracellular transport of cargo. Prevention of local programmed death mechanisms leads to viral shedding. Various viruses show comparable infection strategies [65], such as extending particular stages of the cell cycle, managing programmed cell death, and using the nuclear membrane to transmit viral genetic material to and from the nucleus. These findings help to understand how SARS-CoV-2 can spread via cell-to-cell transmission [65], where ACE2 is not required. Our assessment shows that viral mutations shared by different variants are unsuitable for evaluating disease mechanisms. This is due to the high metabolic interference capacity of the remaining information package of the virus. Attention to mutations in the spike protein has distracted us from evaluating the molecular mechanisms underlying the metabolic dysregulations induced by the virus.

### 3.10. Cluster Analysis

Cluster analysis allows us to extract protein interaction sub-networks that interact with each other in functional complexes and pathways to produce reliable hypotheses that can explain the various dysregulations of human metabolism induced by ORF7b. This also increases the likelihood of identifying candidate genes/proteins that can help us understand the rationale for viral action and the metabolic pathways involved.

Cluster analysis is a data analysis that explores the groups present within a dataset, known as clusters. We used Cluster K-means analysis, which does not need to group data points into predefined groups and is an unsupervised learning [119] method. In unsupervised learning, insights come from the data without predefined labels or classes. K-means is also an iterative partition algorithm, which is a good clustering algorithm that ensures high similarity within clusters and low similarity between clusters. The clusters representing our entire population of interacting molecules in the ORF7b interactome derive from a base of significant experimental data and rigorous procedures for implementing the network. This should produce high-quality clusters, which provide non-redundant and low-noise results as they can reduce the quality and interpretability of the clusters. Assigning a value to K is one of the major drawbacks of this algorithm. In our analysis, K equals 10 (Figure 7).

Except for clusters 1 and 9, which have distinctive features and require separate treatment, the most crucial parametric information is next to each cluster.

This result, which was obtained after many attempts with lower K values, has to be considered as the best compromise. We used this K-value because it gave us the most compact clusters and significant *p*-values (all *p*-values are always <1.0 × 10^−16^). The ten metabolic modules are all functionally consistent, and in Appendix A, we also show the links existing between the clusters. The numerous metabolic relationships existing between the clusters, as shown in the figure, represent the normal metabolic machinery necessary for cellular life. Only the GCC shows an overlay of two modules, but, as we will see, they resolve into two independent sub-graphs. The greatest interest is in these two sub-graphs because they contain most of the identified HUB and bottleneck nodes and control crucial metabolic pathways. The other sub-graphs seem to regulate typical metabolic activities; thus, understanding the specific functions of these central modules and where their constituent proteins operate within the cell is essential. This is a core–periphery organization. Core–periphery is a characteristic we can find at group-level relationships in biological networks, but not exclusively [120]. The situation involves meso-scale dominance events [121]. It describes a scenario where a group of core nodes captures an excessive number of contacts in the network. The peripheral nodes have fewer interconnections with each other, although their sub-graphs connect to the central core. In networking, the mesoscale describes sub-cellular events on length scales ranging from that of a single cell, up to the size of molecular complexes, where groups of molecules self-organize to form large, functional core structures [122]. While individual nodes perform only local operations, their organization into clusters generates a richer and more diverse functional repertoire.

### 3.11. Analysis of GCC Core

From the compact GCC area, the cluster analysis extracted two clusters (1 and 9), both of which are significant and compact. Figure 8 shows cluster No. 1. The caption shows the major topological parameters. This cluster is very compact. Its major role is to regulate the EGFR family signaling pathway (EGFR, ERBB, ERBB2) where the receptors’ protein tyrosine kinase signaling show *p* < 6.85 × 10^−48^. It regulates the Jak-Stat pathway, ERBB and ERBB2 signaling (*p* < 2.55 × 10^−40^), and the regulation of peptidyl-tyrosine (*p* < 2.99 × 10^−27^). We can find the key details in the following GO terms: GO:0007169, GO:0038127, and GO:1901184. But in cluster No. 1, we also find ITGB1, CAV1, EGF, EGFR, PIK3CA, INS, GRB2, PRKCA, HRAS, and MTOR, just to mention the major nodes. Thus, the role of this cluster is also to control cell migration, cell motility, immune response, phosphorylation, cell death, apoptotic cell processes, cell adhesion, cell migration, stress, the insulin path, phagocytosis, lymphocyte activation, blood coagulation, and the cytokine-mediated signaling pathway with very high statistical significance, as it appears from the list calculated by STRING in the biological process (GO) category.

Proteins operate in their specific environments; therefore, knowledge of where proteins are located is crucial to understanding the metabolic processes of which they are a part. We can perform this analysis with the help of Cytoscape. After transferring cluster 1 to Cytoscape, with the help of the STRING app and Node Table (compartment analysis), we selected the protein nodes with the highest statistical value (5.0) that operate in the various cellular compartments. Level 5 collects the most important proteins in defining the biological processes of which they are part.

In Table 12, we see the cellular compartment in which cluster No. 1 proteins operate, and we also see that there are various proteins already defined as dysregulated, so we can know where they operate.
biomolecules-14-00541-t012_Table 12Table 12Operational cellular compartments of cluster No. 1 proteins.CompartmentProteins *Protein NumberExtracellularAREG, **BTC**, **CD81**, CD9, EGF, **EGFR**, **ERBB3**, **EREG**, **HBEGF**, **HSPA8**, INS, LAMA1, LAMB1, **MUC1**, ***NRG1***, **NRG3**, PLAU, SFN, **TGFA**, **TSG101**20Cytoskeleton***CTNNA1***, ***CTNNB1***, **GNAI1**, **GNAI3**, ***LMNA***, **MAPK1**, **PPP2R1A**, **PTPN3**8Plasma membraneADAM17, **ARF4**, **BTC**, ***CAV1***, **CAV2**, CD44, **CD81**, CD82, **CDH1**, ***CTNNA1***, ***CTNNB1***, EDNRA, **EGFR**, EPS15, **ERBB2**, **ERBB2IP**, **ERBB3**, **ERBB4**, **EREG**, GAB2, **GNAI1**, **GNAI3**, **HBEGF**, **HCK**, **HRAS**, ITGA3, *ITGB1*, ITGB4, JUP, KRAS, **LAPTM4B**, **LPAR1**, LPAR3, **LYN**, **MUC1**, ***NRG1***, **NRG3**, PDGFRA, **PIK3C2B**, **PLCG1**, PLCG2, **PPP2R1A**, **PRKCA**, **PRKCB**, **PTPN2**, **PTPN3**, PTPRK, **PTRF**, SHC1, SLC9A1, SLC9A3R1, **TGFA**, **TSG101**, **USP8**54Cytosol**PIK3C2B**, GRB7, **ARF4**, **PPP2R1A**, **PLCG1**, **HCK**, **USP8**, **PRKCA**, **MAPK1**, **RAB5A**, **FOS**, **HSPA8**, ***CTNNB1***, **HIF1A**, GAPDH15Mitochondrion**PPP2R1A**, **MAPK1**, HSP90AA1, ***LGALS3***, **ERBB4**, **PTRF**, MT-CO27Golgi**CAV2**, CBL, **CDH1**, **HRAS**, **LYN**, **MAPK1**5Endoplasmic Reticulum**FOS**, **NCK1**, **PTPN2**3PeroxisomeNo level 5 protein-Endosome**CDH1**, **EGFR**, ***CAV1***, **ERBB2**, PTPN1, **RAB5A**, **MAPK1**, **TSG101**, **GRB2**, HGS, **USP8**, **LPAR1**, **LAPTM4B**, **GRAP2**14Lysosome**LAPTM4B**, **HSPA8**, MTOR, **HCK**4Nucleus**CAV2**, ***CTNNB1***, **EGFR**, **ERBB2**, **ERBB2IP**, **ERBB4**, **FOS**, **GRAP2**, **GRB2**, **HIF1A**, **HRAS**, **HSPA8**, IGFBP3, JAK2, ***LGALS3***, ***LMNA***, **LYN**, **MAPK1**, **MUC1**, **NCK1**, NCL, ***NRG1***, **PLCG1**, **PPP2R1A**, **PRKCB**, PRKDC, PTPN11, **PTPN2**, PTPN6, **PTRF**, *STAT1*, STAT3, STAT5B, TFAP2C34Note: (*) Only proteins with the highest statistical significance value of 5, according to Cytoscape (values range from 0 to 5). We calculated protein node compartmentalization and values in Cytoscape using the STRING app. Highlighting all the proteins in Table 9 would have made this table unreadable. (1) Bold black font indicates the proteins that are present in more than one compartment. (2) We identified and underlined the proteins responsible for the dysregulation of ERBB signaling, EGFR, protein tyrosine kinase activity, and regulation of peptidyl-tyrosine phosphorylation, as shown in Table 9. (3) The proteins that are also involved in deregulating apoptosis and anoikis to allow for diffusion are indicated in red italic font. Proteins in bold black font and underlined or in red italic and bold font are common to two groups.


The nucleus and plasma membrane, as well as the cytoskeleton, are among the richest compartments of functional activities where proteins crucial for the progression of these activities operate. In Table 11, we find many of these proteins, for which symbolic notations are used to distinguish them (see the note of the Table 12). The table summarizes two important proteomic characteristics as follows: (a) there are numerous proteins that operate in a multipolar way, i.e., in several compartments (e.g., EGFR) and (b) there are many dysregulated proteins, in particular, those involved in the fundamental processes of signaling and in favoring cell diffusion. Various proteins localize in multiple compartments, showing a shared protein pool even if unrelated. However, each protein has its own level of expression and its own compartmental distribution. Taking into consideration the entire picture, we can interpret this as a sign of functional advancements starting from the membrane and progressing toward the nucleus. The limit is the absence of temporal information that flattens the metabolic dynamics and makes it very difficult to make reliable sequential explanations. But this is not the only intricacy. Appendix A demonstrate how single nodes can take on multiple roles to engage in various functional processes. Even a single functional activity can have its nodes distributed in many modules. This is a straightforward demonstration of how difficult it is to describe the actual behavior of concurrent functional processes without a temporal chronology, but the entire network, i.e., the operational context, can help.

Not only the regulation of space and time but also the compartmentalization characterizes the cellular proteomes. The presence of similar proteins in different compartments indicates the existence of distinct local proteomes [123], each carrying out specific metabolic activities, making it challenging to detect any distortion. The nucleus and cytoplasm are among the most populated compartments. The proteomes of these compartments show a multipolar protein distribution, which makes them very ductile functionally. Therefore, attributing static and specific roles to metabolism and to the proteins that operate within is a process that does not correspond to reality. We cannot attribute a protein’s metabolic function to its presence or absence. Its function is also determined by the reactions that happen at different omic levels and compartments [124] and reactions that are always the result of protein–protein interactions. Thus, the interactomic level reflects what happens at the genomic or transcriptomic level, generating a network that differs from the underlying ones. The event in question has gained prominence recently [125]. Some melanoma cells show a dependence on external sources of methionine for their growth. The authors describe the methylome, transcriptome, and proteome of these cells. Only the multilevel contemporary study allowed the authors to understand the real metabolic behavior of methionine addiction because the study of the methylome alone led to trivial conclusions.

In short, we obtained the spatial distribution of proteins in the ORF7b interactome, but the temporal distribution is missing. Multi-localization of a protein increases the probability of interactions, generating possible new functional characteristics in the context. This expands the functional capabilities of the cell but makes any modeling that does not include all the parameters involved difficult.

Because of important proteins, cluster No. 9 has the potential to perform multiple functions (Figure 9). This cluster controls the process that modulates the cell transport to, or maintained in, a specific location (GO:0032879, *p* = 2.30 × 10^−34^); the extent of the addition of phosphate groups to a molecule (GO:0042327, *p* = 1.89 × 10^−29^); cell migration (GO:0030334, *p* = 3.11 × 10^−29^); regulation of cell migration (GO:0030334, *p* = 3.11 × 10^−29^); and the transmembrane receptor protein tyrosine kinase signaling pathway (GO:0007169, *p* = 1.78 × 10^−28^). It is also associated with the negative regulation of cell death (GO:0060548, strength = 0.92, *p* = 1.75 × 10^−17^) and programmed cell death (GO:0043069, Str. = 0.90, *p* = 3.96 × 10^−16^) and in the negative regulation of the production of miRNAs involved in gene silencing (GO:1903799, Str. 1.78, *p* = 4.6 × 10^−4^). Similar considerations also apply to cluster No. 9.

Dysregulated proteins, such as CTNNB1, SRC, PTK2, ITCB3, or PRKCD, found in cluster No. 9 (see Table 13), are present in many cellular compartments, including those that are distant from each other or different from a chemical–physical point of view, such as the cytosol and plasma membranes. This means that they regulate their expression over time and that they require post-translational modifications that depend on the context. Most analysis platforms collapse this information onto the native protein, so nodes have more functional connections than context. This induces errors in the degree value and the related topological evaluations, which can lead to alterations in the network.

An instance of this is the activation of the Human SRC (P12931, Proto-oncogene tyrosine-protein kinase Src), a non-receptor protein tyrosine kinase that is triggered upon binding to various cellular receptors, including integrins and other adhesion receptors, regulating a wide range of biological processes. It belongs to the Src kinase family and is functionally redundant, making it challenging to identify its specific role in each compartment and determine which member is involved without the knowledge of its spatio-temporal characteristics in that specific context.

### 3.12. Co-Regulation between Hub and Bottleneck Proteins, Transcription Factors, and miRNAs

Our findings thus far revealed a metabolic depiction that outlines the involvement of a specific group of significant high-ranking proteins in a series of dysregulated metabolic processes aimed at promoting the dissemination and spread of virus-infected cells throughout the body because of the influence of the accessory viral protein, ORF7b. However, we still have limited vision because we can only glimpse at the purposes and know some of the involved actors, but we still cannot understand which actors planned and performed the entire process.

Gaining insight into the intracellular mechanism of complex biological processes driven by ORF7b also hinges on deciphering its intricate co-regulatory network. Identifying hub and bottleneck proteins in protein groups dysregulated by viral infection prompts investigation to understand their co-regulation. Within the co-regulatory network, there are both post-transcriptional and transcriptional regulators that can regulate themselves and each other.

A limitation that should give pause for thought is the evidence that hub and bottleneck proteins control and regulate an enormous number of functional processes. Discovering their involvement in a particular process does not necessarily indicate the reality and existence of that process. Precise rules govern a functional process, primarily depending on the context of the events and the chemical–physical characteristics of the compartmentalized microenvironment where the event should occur. To ensure a functional event, the cell must program when, where, and how it should occur. The metabolic network is not solely dependent on proteins. To synchronize basic functional activities according to the circadian cycle or unexpected events, several other actors are needed to accelerate or slow down an intricate and dynamic system. The comprehension of co-regulatory mechanisms that are fundamental to cellular identity and function requires the involvement of transcription factors (TFs) and microRNAs (miRNAs). TFs and miRNAs work together to regulate transcription and post-transcriptional processes [125,126].

The integration of computational and experimental interaction data in network models has the potential to emphasize functional mechanisms in TF- and miRNA-mediated gene regulation. These models can provide insight into the mechanisms that control gene expression at the system level rather than at the individual gene level. Typically, TFs act as activators or repressors, increasing or decreasing transcription, while miRNAs are repressors. We can visualize the distinct activities by using two separate networks as follows: transcriptional networks and post-transcriptional networks. It is noteworthy that both networks are bipartite and direct. In each network, there are two distinct types of nodes interconnected by unidirectional edges. One network contains interactions between genes and transcription factors, which is known as a transcriptional regulatory network. The other network contains interactions between genes and miRNAs, which is known as a post-transcriptional regulatory network. We assume that in post-transcriptional regulations, the regulatory actions of miRNAs toward targets are negative. However, it is possible to obtain integrated gene regulatory networks that include genes, TFs, and miRNAs, provided that the components are statistically more significant. The databases on TFs and miRNAs are quite recent, and the data collected are both experimental and predictive because this area of research is still very young. Selective filtering is required to obtain significant nodes. As reported in Section 2, the reference databases of transcriptional and post-transcriptional networks comprise experimental data, whereas the integrated co-regulatory database comprises mixed data. This means that the comparison of the integrated co-regulatory network with the transcriptional networks may yield diverse interactions, which depend on the respective node rank in the two distinct systems.

### 3.13. Transcriptional and Post-Transcriptional Regulatory Networks

As a result, in transcriptional regulatory networks, TFs possess two types of action since it is the TF that binds to its target gene rather than the reverse. The information comprises an in-degree, which signifies the number of transcription factors binding a gene, and an out-degree, which signifies the number of genes bound by a transcription factor. All this reflects the functional and biological aspects underlying these interactions. High-grade TFs (i.e., hub TFs interacting on many genes) have a high key character of biological functionality, while target genes bound by many TFs do not have a tendency to be essential functionally. Therefore, analyzing this type of network provides insights into biological systems that are not obtainable through single-gene studies. Figure 10 represents both the networks containing TFs and miRNAs, illustrating the transcriptional and post-transcriptional networks of gene interactions, which include hubs and bottlenecks. The transcriptional network reveals that EGFR, the top-ranking hub node within the PPI network, possesses an in-degree value of 1 in relation to its interaction with ZNF263, whereas ZNF263 exhibits an out-degree value of 3. Therefore, within this network, ZNF263 holds greater biological significance in relation to EGFR. Its role in this transcriptional network involves functioning as a DNA-binding transcriptional repressor that targets RNA polymerase II, resulting in the repression of EGFR, PIK3R1, and VAMP2. The TFs and miRNAs represented in the two networks are those of higher rank with a higher probability of interaction.

### 3.14. Co-Regulatory Network

Establishing a co-regulated network involves the integration of HUBs and bottlenecks with FTs and miRNAs. To determine the transcriptional regulatory relationships that these nodes may hold, we employed hub and bottleneck as enrichment seeds. This co-regulated network allowed us to pinpoint the 14 most reliable TFs and two miRNAs that were associated with the expression of HUB and bottleneck genes.

The network (Figure 11) shows that among bottlenecks, SEC13 is one of the most regulated genes. The protein encoded by this gene belongs to the SEC13 family of WD-repeat proteins and is a component of several important complexes. It is a component of the nuclear pore complex (NPC), which regulates transport between the nucleus and cytoplasm and has a direct role in regulating gene expression [127]. It is also a component of the COPII Coat Complex, where it plays a role in coated vesicles [128]. Four of the transcription factors that regulate SEC13 also regulate PIK3R1, the gene responsible for encoding Human_P85A, a protein that modulates glucose uptake in insulin-sensitive tissues by binding to activated Tyr kinases on the cellular membrane. Because of its inhibitory action, it appears to be a significant factor contributing to the hyperglycemia observed in patients with COVID-19. EGFR is also controlled by several TFs. Governing each of these genes is multifaceted and bolstered by two miRNAs, hsa-miR-576-5p and hsa-miR-1. The role of miRNA expression levels in disease processes and physiological development is significant, as changes in microRNA copy number or expression are associated with the onset of various human diseases [129]. miRNAs are present at substantial numbers in humans [130].

The correlation between miRNAs and human genes during SARS-CoV-2 infection is still an expanding research field including only initial studies. Some preliminary evidence shows potential associations between miRNAs and genes that participate in the reaction to infection. It is essential to highlight that the analysis of this subject is still in progress. Despite this, there is ongoing research analyzing the potential association between miRNAs and genes in the response to SARS-CoV-2 infection to regulate inflammation. miRNA-155 [131] links the regulation of genes involved in inflammation, such as tumor necrosis factor-alpha (TNF-α) and interleukin-6 (IL-6). Based on earlier research on COVID-19, miRNA-146a might play a role in controlling the innate immune response [132], and an increase in its expression could contribute to the disruption of inflammatory pathways. miRNAs might exert direct control over the replication of SARS2, as well as its capacity to infect host cells [133]. This could involve both regulating viral proteins and genes/proteins involved in human metabolism. Observations in cell lines and cancer patients led researchers to predict that miR-576-5p could down-regulate both PIK3CA and its mRNA [134]. Meanwhile, their target mRNAs were up-regulated. Researchers have linked hsa-miR-1 to the regulation of human genes, especially in cancer patients [135]. It has been observed that this specific miRNA contributes to disturbing glycemia for individuals with type 2 diabetes [136].

The co-regulatory network provides a better picture of metabolic events that the simple identification of a gene or protein in a metabolic pathway cannot provide, even more so when we study the molecular mechanisms involved in pathology. Stating the involvement of a protein or gene in a pathological state without fully grasping the coordinated activity of genes, miRNAs, TFs, mRNAs, and proteins may not always lead to accurate conclusions. Co-regulatory networks offer a more decisive direction by elucidating the general coordination of the aforementioned actors, besides the appraisal of the pathological consequences of ORF7b.

### 3.15. Comparative Analysis of Negative Regulations according to GO

Figure 12 shows the set of negative regulations vital for cellular diffusion represented by three transcriptional networks, which, upon comparison, exhibit remarkable similarities. In all three networks, EGFR, HRAS, HSPA5, PIK3CA, PIK3R1, and SRC are the genes involved in the negative control of programmed death. At the individual gene level, DNA-dependent transcription exerts negative control over their transcription.

Below is a brief illustration of the most intriguing transcription factors found in the networks. ZNF423 and ZNF263 (Zinc Finger Proteins 423 and 263) can act as both transcriptional repressors and activators by binding to DNA, where ZNF423 plays a central role. MXD4 (Max Dimerization Protein 4) is a transcriptional repressor complex. PHF8 (histone lysine demethylase PHF8) is a transcription activator that acts on the epigenetically methylated histone 3 but is a repressor for the methylated histone 4. It acts as a coactivator of rDNA transcription by activating polymerase I (pol I)-mediated transcription of rRNA genes and playing a role in the cell cycle. However, its role is still unknown in vivo. GABPA (GA Binding Protein Transcription Factor Subunit Alpha) is a transcription factor that interacts with purine-rich repeats (GA repeats), so it positively regulates the transcription of transcriptional repressor RHIT and the ZNF family such as ZNF205. MLX (MAX Dimerization Protein MLX), its decoded product (Max-like protein X), forms many sequence-specific DNA-binding protein complexes with various proteins. These complexes act as transcriptional repressors. It plays a peculiar role as a transcriptional activator of glycolytic target genes; thus, it is involved in glucose-responsive gene regulation. Here, we have another pro-glycemic effect that is common to patients with COVID-19.

While Figure 13 shows the relationships between genes and miRNAs involved in blocking programmed death at the post-transcriptional level, Figure 14 shows its co-regulated network, where we find Myc and TP53, two well-known transcription factors. MYC, (MYC Proto-Oncogene or BHLH Transcription Factor, which codes for P01106 · MYC_HUMAN) is involved in many diseases [137]. The Gene Ontology (GO) annotations that concern MYC comprise DNA-binding transcription factor activity and the ability to function with TAF6L to activate target gene expression through RNA polymerase II cis-regulatory region sequence-specific DNA binding.

TP53, also known as Tumor Protein 53 and encoding for P04637, cellular tumor antigen p53, acts as a tumor suppressor in response to cellular stresses to regulate the expression of target genes [138]. However, in specific metabolic contexts, it can induce cell cycle arrest, apoptosis, and changes in metabolism [139]. In fact, researchers have discovered that SARS-CoV-2 infection leads to stabilizing TP53 on chromatin [140], which contributes to a robust host cytopathic effect. The participation of this protein results in the alteration of chromatin accessibility, cellular senescence, and the release of inflammatory cytokines via TP53 in response to various SARS-CoV-2 spike variant-induced syncytia formations.

The protein appears to have a role in inflammation associated with cellular senescence [140]. In addition, researchers discovered that TP53 plays a role in IFN-γ-mediated signaling, apoptosis, and proteasomal degradation of CD4 T cells [141]. However, uncertainties regarding the functionality of miRNAs persist because of technical difficulties and the considerable number of miRNAs that are still subject to systematic profiling [130]. Because they have low intrinsic stability and contain RNAses [142], their measurements can be compromised by degradation and the effects of laboratory manipulations [143,144].

TFs are well-established proteins with reliable experimental results, although miRNAs remain somewhat enigmatic. TFs are proteins that control the rate of transcription of genetic information from DNA to mRNA binding to DNA. Thus, their function is to regulate genes by switching them on and off. This functional activity addresses gene expression to the exact target cells at the right time and in the right amount. Groups of TFs function in a coordinated fashion to direct cell division, cell growth, and cell death. TFs work alone or with other proteins in a complex by promoting (as an activator) or blocking (as a repressor) the recruitment of RNA polymerase to specific genes.

We examined the various correlations between miRNAs, TFs, and the components of the compact hub-and-spoke architectural system of the PPI network, obtaining information on the fundamental co-regulations operated by some TFs and miRNAs. These findings suggest that crosstalk motifs, comprising the direct and non-shared relationships between regulators and their target genes, can have downstream effects on diverse biological processes, in line with the features already highlighted in the interactome’s analysis network. This analysis amplifies and substantiates our findings and deductions from the interactomic analysis. Our result, however, has limitations. The human genome contains thousands of coding and non-coding RNA genes. These genes are expressed differentially, in diverse locations, at distinct times during normal homeostasis, or in response to environmental cues. This differential expression also extends to TFs and miRNAs. Gene regulation is specific to certain conditions and changes over time, meaning our findings only provide a static view of the molecular mechanisms affected by ORF7b. While our conclusions are valid, we can only show the presumed targets, not how they work dynamically.

## 4. Discussion

The guiding principle that underpins this research is that SARS-CoV-2 infection leads to changes in the deep metabolic activities of infected cells to favor the acquisition and maintenance of viral strategies compared with normal cells. The virus causes a reprogramming of cellular metabolism by its proteins, where “metabolic reprogramming” refers to the recognition of normal metabolic pathways that are modified by viral proteins when compared with those in normal tissue. This point is significant because the analysis of “metabolic normality” is often overlooked. Our training in cancer has taught us to search for mutations that can modify signaling processes. In viral infections, mutations are absent, as viruses achieve the same aim by up- or down-regulating normal signaling pathways or other metabolic processes.

Our results reveal the functional impact of the accessory protein ORF7b in SARS-CoV-2 infection and identify molecules that control metabolic processes dysregulated by this viral protein. Among the many functional activities highlighted, we focused on those that promote the spread of infected cells in the organism.

The release of virions into the extracellular space is a common event among many viruses, which has stimulated the study of virus egress/entry biology. Although some viruses spread through infected cells [145,146], this is an understudied topic. Several authors [65,147] have recently reported evidence from antibody experiments that SARS-CoV-2 could spread through cell-to-cell transmission. However, no one has studied or hypothesized any related molecular mechanism. In this article, we confirm those authors’ hypotheses and describe the deep molecular mechanism that underlies this feature of SARS-CoV-2. This discovery contributes to our understanding of the human immune response to the attack of this virus because cell-to-cell transmission is an effective means by which viruses evade host immunity.

It should be considered that our data on spread, in some sense, support the theory of virulence evolution, which assumes that high growth rates of pathogens should both increase transmission between hosts and increase disease-induced morbidity or mortality [105,148]. This logic accommodates the spread of infected cells, but the theory also suggests that viral “tolerance” mitigates virulence without reducing viral load [148,149]. This also suggests that the host should select the pathogen with a higher growth rate to gain a gain in transmission between hosts but without being detrimental to the original host [149,150,151]. Today’s clinical data tells us that the virulence of COVID-19 is decreasing with no type of specific intervention. Our data do not explain the effects of diffusion on the virulence, but they pave the way for experimental designs with greater awareness of what happens.

Using interactomics, we analyzed only functional and physical correlations between ORF7b and the entire human proteome determined by experiments. To obtain reliable interactomics results, from the set of interactors, we extracted only those that were characterized by high significance. The investigation showed that the virus achieves its strategic goals by interacting with metabolic processes controlled by human proteins such as EGFR, SRC, HSPA5, MTOR, SEC13, SEC61A1, VAMP2, PIK3R1, PIK3CA, GRB2, and HRAS, which are important for human metabolism because they are high-ranking HUB and bottleneck proteins. Through a series of analyses using transcriptional co-regulation networks, we also validated our results by identifying regulatory actions conducted by transcription factors and miRNAs on genes that code for the identified key proteins.

Viruses do not perform metabolic processes but know how to interact with them to their own advantage. Although researchers have made various attempts to identify metabolic pathways and nodes under the control of the virus, as far as we know, this is the first wide-ranging interactome map identified for a specific protein of SARS-CoV-2. We identified some metabolic pathways under the control of ORF7b; nevertheless, we have limited knowledge of the comprehensive set of viral proteins involved and the specific mechanisms. Despite that, several authors have hypothesized some functional activities of ORF7b in the infected cell and its synergism with other viral proteins, but no one has attempted to study in depth the molecular and functional interactions within human metabolism implemented by ORF7b. In particular, its involvement in SNARE-driven vesicular transport, exocytic processes, and ERBB signaling has been identified but without a functional characterization that identifies the actual role of ORF7b in synergy with other viral proteins [152]. There are other studies that have endeavored to juxtapose the mechanisms of diseases between SARS1 and SARS2 [153,154], but none of them have deciphered any common molecular mechanism. Both viruses lead to acute respiratory distress, but many phenomenological observations show differences. One study predicted that SARS-CoV-2 induces a systemic disease, which, unlike SARS1, damages various organs in the body, such as the heart, kidney, and brain [155,156]. These results suggest that the two viruses use different molecular mechanisms, but we do not know which mechanisms they use. Out of curiosity, searching PubMed for “differences in molecular mechanisms of SARS-CoV-1 and SARS-CoV-2” or “molecular mechanisms of SARS-CoV-1 and SARS-C0V-2 (or similar terms)” yielded no results. The continuous work conducted by the curators of BioGRID, in selecting and evaluating the statistical significance of each single experimentally characterized interaction between the viral proteins and the human proteome, has allowed us to design this study with the methods of interactomics. Direct knowledge of the deep molecular mechanisms implemented by individual viral proteins is essential because only through this knowledge will we be able to design specific and effective antiviral drugs. Conducting a study at a deep molecular level, which is a research area still rather obscure in its modes of action in space and time, is an important approach aimed at identifying those human proteins that play crucial roles in viral infection such as hub nodes or as a bottleneck. These proteins represent the crossroads of multiple biological activities and, therefore, are the best targets for disease control.

ORF7b is a tiny viral protein of 43 amino acids, a macro-polyanion with a net charge of −4 at neutral pH (four negative residues and no positive charge); the central part from 9 to 29 is helical, and the protein surface is negative [6]. This protein does not appear to operate on its own (see Table 9). What emerges from this study is the precise interference of ORF7b with various molecular mechanisms at the basis of our metabolism. ORF7b showed diverse behaviors, in terms of localization, membrane recruitment, and metabolic dynamics. The results show its important role in conditioning cellular transport processes as well as in some important signaling pathways (see Table 3). The topological characteristics of this interactome reveal a group of proteins with structural and functional properties that are implicated in multiple metabolic activities, some of which are dysregulated by ORF7b’s action. Their high functional relationships characterize these proteins and their high ability to regulate a multitude of significant metabolic and signaling pathways. The interactome shows certain metabolic modules that perform necessary functional activities for normal cellular metabolism. A large central core (GCC) comprising two connected clusters was identified through cluster analysis as the primary functional location of these proteins. The high number of tight connections favors a high metabolic rate, which accelerates any functional activity.

The activity of these proteins extends to very different places in the cell (see Table 4) according to a hub-and-spoke topological model and also to those tissues that have molecular characteristics suitable for the entry of the virus (see Table 6). All this suggests that ORF7b must have a remarkable ability to interact with different molecular partners, allowing it to operate everywhere, at the membrane level and in the cytoplasm. Indeed, the list of its main molecular interactors shows both membrane proteins and cytoplasmic proteins. Some authors have hypothesized a role for ORF7b as an intrinsic single-span membrane protein, in analogy with the 44-amino acid homolog ORF7b of SARS [6]. This hypothesis is rather restrictive considering the wide spectrum of functions in which this protein is involved and the spatio-temporal characteristics that a biological object of this type must possess in order to be involved in various intracellular transport processes (see GO:0006810, *p*-value 3.04 × 10^−67^) or even in guiding and regulating target localization (see GO:0008104 and GO:0045184, with *p*-values of 1.4 × 10^−58^ and 2.85 × 10^−58^). But this protein must also have the ability to interact with different membrane systems (see GOCC:0016020, GOCC:0031090, GOCC:0031982 or GOCC:0098588, with *p*-values of 2.5 × 10^−92^, 2.07 × 10^−77^, 5.13 × 10^−62^ and 3.17 × 10^−58^) and to interfere with metabolic signaling paths (see GO:0007169, GO:0007167; HAS-9006934, HAS-1227986, or HAS-6811558, with *p*-values of 1.23 × 10^−66^, 7.95 × 10^−59^, 4.44 × 10^−84^, 2.43 × 10^−30^, and 5.16 × 10^−24^).

The regulated functional re-localization seems to be one of the most important characteristics of this protein [157]. ORF7b shows coherent functional solutions with viable biochemical functional models. The closest class of proteins possessing these types of broad properties is called the “Peripheral Membrane Proteins”, which is a class of proteins that live at the membrane interface [158,159]. In 2002, Felix Goñi [160] introduced the concept of “non-permanent membrane proteins” to encompass the wide variety of proteins that are not found in a stable membrane-bound form under physiological conditions but interact with the membrane in certain phases of their specific course of action. Despite the fundamental biological meaning of these proteins, an experimental characterization of their structure has always been vague because attempts at structure prediction often fail. Therefore, this protein class has a poor representation of its 3D structures within the PDB because they are difficult to study [161]. Representing them in a few words, they are soluble proteins that bind transiently to the surface of biological membranes or even to proteins on the outer side of the membrane where they perform their functions. The reversible attachment of proteins to biological membranes shows how they can regulate cell signaling and many other important cellular events through a variety of mechanisms [162,163]. Thus, the behavior of peripheral proteins, reversibly associated with the lipid bilayer [162,164], may also explain the behavior of ORF7b, consistent with its structural/functional properties. Therefore, this protein can be considered a reliable member of the class of “non-permanent membrane proteins” [165].

Recent molecular dynamics simulation experiments provided molecular insights into the protein’s dimerization [166]. This cited study shows different dimerization models, both parallel and antiparallel. Among the various structures modeled, the authors suggest the possibility that the parallel dimer may operate docked to the membrane, from 7 to 30, and float in the cytoplasm from 31 to 43. Based on their results, they also observe that the analysis of genetic mutations of ORF7b during the evolution of the pandemic showed a loss of stability in the homodimer not compatible with the molecular mechanisms that regulate the production of IFN, the functional activity for which this protein is most credited. They conclude that the lack of detailed structural information on lateral protein–protein associations hinders a thorough evaluation of packing, so there is not yet sufficient detail to define consistent structure–function relationships. This information adds to the previous considerations, but every hypothesis made remains valid.

Like other viruses, SARS-CoV-2 can cause reinfection/reactivation and persistent infection, as supported by several experimental studies [167]. SARS-CoV-2 has the potential to activate or modulate oncogenic cancer-promoted pathways, leading to chronic low-grade inflammation and tissue damage, according to growing evidence [168]. Several authors perceive oncogenesis as a potential long-term effect of SARS-CoV-2 infection, which could lead to the onset of cancer by inhibiting tumor suppressor genes [169]. Utilizing similar tactics as EBV or HSV1 by SARS-CoV-2 to manipulate p53 is clear, as the virus takes over the protein using viral antigens, which leads to p53 deterioration [170]. By deactivating both the external and internal apoptotic pathways of host cells, SARS-CoV-2 may spread like cancer cells [171,172]. Our results suggest that the cancer-like effects of SARS-CoV-2 result from the virus’s capability to spread infected cells through its proteins, mimicking cancer and its metastasis. The lack of adequate understanding of these mechanisms makes it impossible to make accurate predictions about the long-term implications of long COVID.

However, we should make a last consideration given the recent advances in our understanding of the N protein of SARS-CoV-2. Phosphorylation of the central disordered region of the N protein forms dynamic, liquid-like condensates that also control viral genome transcription [103]. The N protein contains three dynamic disordered regions that house putative transiently helical binding motifs, and the protein undergoes liquid–liquid phase separation [103,173]; thus, phosphorylation regulates the accessibility and assembly of the N protein to bio-condensate [174]. Another critical function of N is to encapsulate the viral genome of ssRNA to evade immune detection and protect viral RNA from degradation by host factors [174,175,176,177].

Viral proteins form condensates for their molecular strategies, such as infection and signaling transduction [178,179]. Viruses execute their molecular tactics in specific parts of cells. For instance, we should consider how phase separation in cell compartments affects important processes like viral transcription or viral spread [178,179]. That ORF7b interacts with N (Table 9) supports the involvement of ORF7b in viral diffusion phenomena, with mechanisms of alteration of the cytoskeleton, but perhaps also with more complex mechanisms involving liquid droplets. After all, phase separation is one of the basic molecular processes that govern multiple cellular activities, such as cancer progression, gene expression, and signaling transduction [180].

In SARS-CoV-2, the properties of the liquid-like condensate that forms phase-separated compartments without a membrane and the transient nature of interactions within them are determined by the interaction between the N protein and viral RNA because of its intrinsic disorder properties [181]. The threshold for phase separation decreases as the number of interacting sites of a molecule increases. This multivalency comes from structural domains, where each domain contributes to binding [181]. Intrinsically disordered regions (IDRs) often participate in phase separation, as they might provide a source of multivalency. In fact, the low affinity of their individual interactions can enable liquid-like properties [181].

We studied SARS-CoV-2–host interactions in a simplistic context because crucial information is missing from the databases. For example, there have been few studies evaluating virus–host molecular interactions considering the range of post-translational modifications. Without the enormous potential of the biological role of post-translational modifications, we run serious risks of having distorted information on the biology of this virus. Researchers have shown the crucial roles of phosphorylation and ubiquitylation in other systems but have not yet identified the corresponding proteoforms in SARS-CoV-2–host interactions.

However, many of the high-ranking proteins we studied and selected show that they have all the characteristics necessary to act even through forms of bio-condensates. Therefore, we cannot exclude that, together with the co-regulation that we highlighted, there may also be a further form of regulatory activity exerted by the liquid-like condensates. We cannot exclude it considering their well-established presence in cells and the important roles they play.

## 5. Conclusions

The proposed model, although created on the most robust basis possible given our current knowledge of the interactions between ORF7b and other SARS-CoV-2 proteins with the human proteome (see “Robustness of the study” in the Appendix A), will be worth seeing again, supported by more precise knowledge on transcriptional modifications and the spatio-temporal characteristics of its proteins and by the role of bio-condensates. Deep biological aspects are still very little known and often overlooked. Without this, we will continue to have inconsistent flat views of metabolism and viral action.

The results were examined in the context of our current understanding of the principles responsible for cellular behavior emerging from an interactomic analysis. At the same time, our work offers a mechanistic hypothesis to explain aspects of the virulence of SARS-CoV-2, demonstrating key differences in using the mesoscopic approach of Systems Biology compared with symptom-based macroscopic approaches, which tells us very little about what might be happening at deep metabolic levels in the human body, as long as it is based on omics data, both experimental and significant, because this is the real limit. Our interactomics framework indeed offers a series of testable questions and predictions that can stimulate future work, such as comparing deep mechanisms of virulence evolution in diverse infection stages. Understanding the molecular mechanisms that select the evolution of viral traits in the human host should allow us to better predict and combat the virulence of probable future threats and also understand the most suitable targets for designing a drug.

## Figures and Tables

**Figure 1 biomolecules-14-00541-f001:**
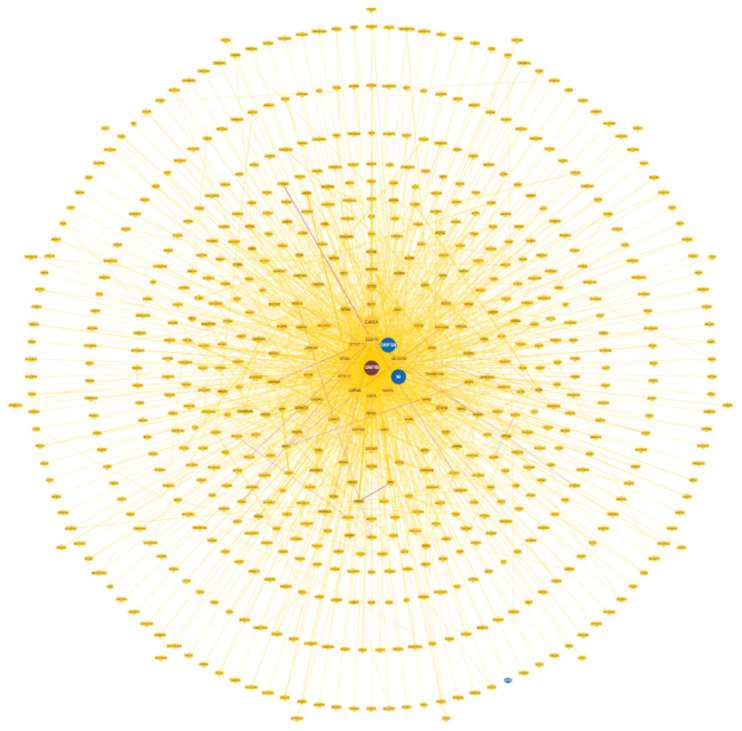
Circular network of SARS-CoV-2-ORF7b and human host PPI (from BioGRID). The circle within circle representation shows the layers closest to the center as more highly connected. BioGRID also suggests the likelihood of direct/indirect interactions between ORF7b (in dark red) and other viral proteins (ORF3a and M, in blue). The proteins used in the present analysis are among those in the most densely represented central area.

**Figure 2 biomolecules-14-00541-f002:**
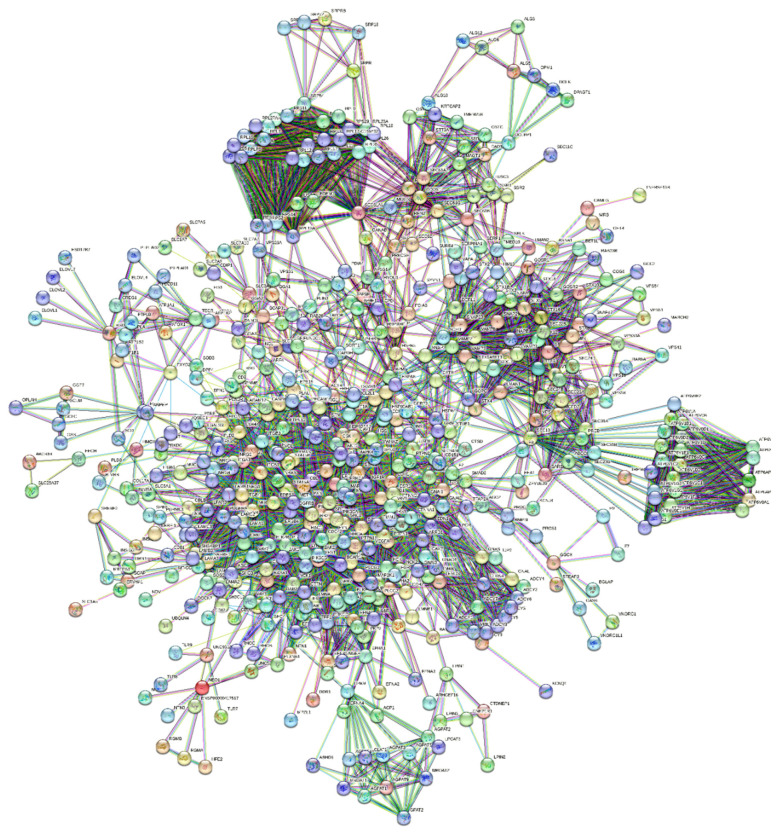
Interactome of 51 human proteins functionally involved with ORF7b2, enriched with 500 first-order proteins. The overall structure reveals peripheral compact groups of nodes that can represent specific functional modules or even particular protein complexes. The network was calculated by STRING, and the score is 0.9. The number of edges is greater than the number of nodes in a similar random network we calculated (PPI enrichment *p*-value < 1.06 × 10^−16^). The topological parameters are shown in Table 1.

**Figure 3 biomolecules-14-00541-f003:**
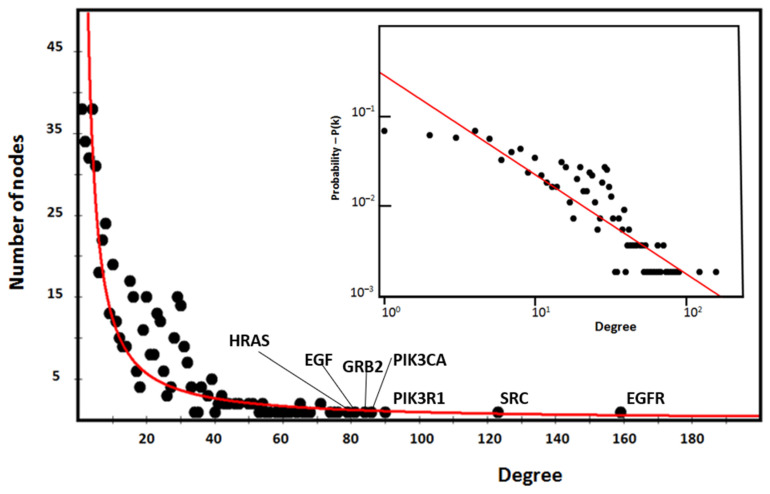
Node Distribution. The distribution follows a free-scale distribution based on the power law. In the inset, we present the same nodes on a log–log scale, with the best fit of data shown in red.

**Figure 4 biomolecules-14-00541-f004:**
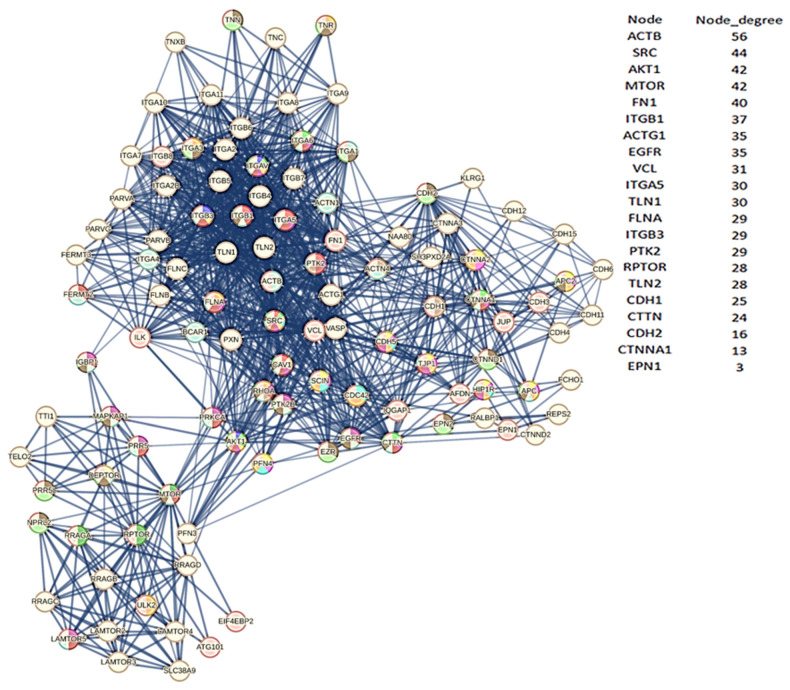
Relationships among cytoskeleton-related proteins. The network (top left side) has a score of 0.7 (high confidence); all seven source channels are active; enrichment of the 8 basic proteins as functional seeds with 100 first-order proteins. Enrichment of up to 100 proteins was necessary to achieve the integration of all eight proteins into the network without expanding the number of functions too much. Topological data: number of nodes, 108; number of edges, 872; average node degree, 16.1; avg. local clustering coefficient, 0.697; enrichment *p*-value < 1.0 ×10^−16^.

**Figure 5 biomolecules-14-00541-f005:**
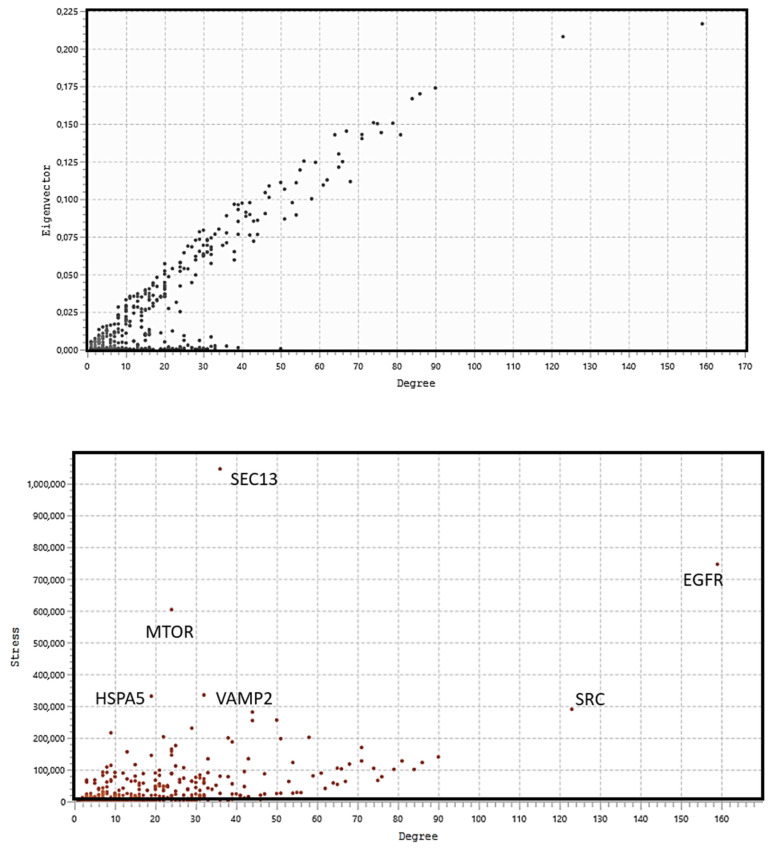
Eigenvector distribution (**top**); stress distribution (**middle**); and betweenness centrality distribution (**bottom**). We calculated distributions using Cytoscape with Analyzer and CentiScaPe. By cross-referencing parametric values, we selected the best proteins in the Cytoscape Node Table for each protein.

**Figure 6 biomolecules-14-00541-f006:**
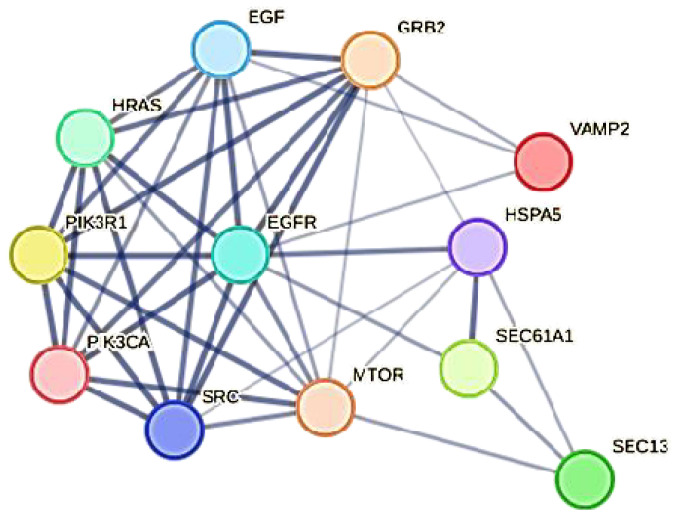
Hub-and-spoke organization of major HUBs in the ORF7b-induced human interactome. By removing all unnecessary nodes from the network in Figure 2, we extracted this graph. Edge intensity is proportional to the interaction intensity between nodes (calculated by STRING).

**Figure 7 biomolecules-14-00541-f007:**
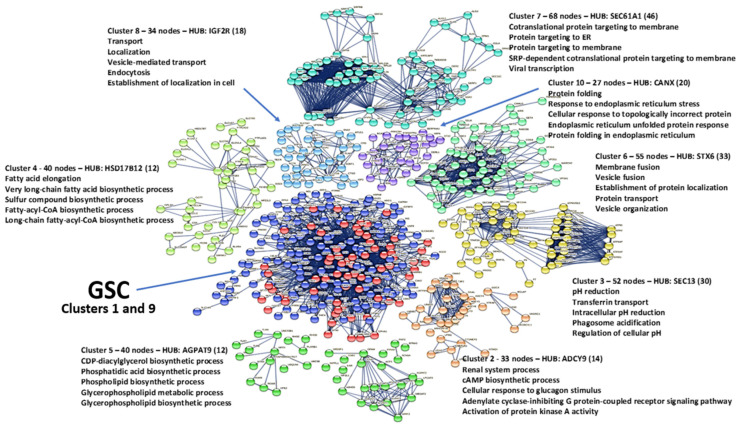
Clustering. The analysis shows ten clusters that are all identifiable except for the two central ones. All 10 clusters are significant with *p*-values < 1.0 × 10^−16^. The number in the brackets next to each key hub indicates its degree. We did not highlight the links between clusters to ensure they are visible. Two overlapping central clusters make up the Giant Connected Component (GCC), totaling 206 nodes, which accounts for 37% of the entire interactome.

**Figure 8 biomolecules-14-00541-f008:**
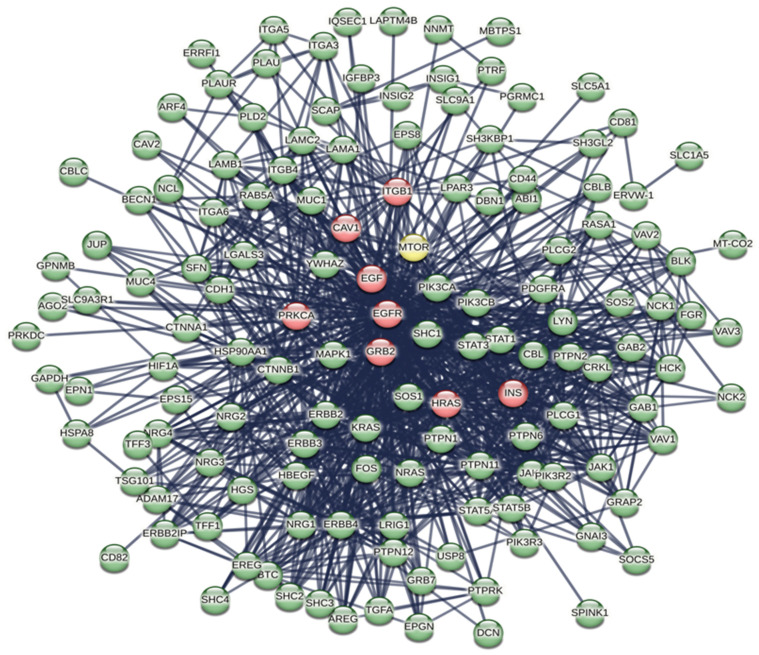
Cluster No. 1–140 nodes, 1110 edges, *p*-value < 1.0 × 10^−16^. Average node degree 15.4, avg. local clustering coefficient 0.622 (expected number of edges in a similar random network, 202), network diameter 3, network radius 2, characteristic path length 1.91, network density 0.108. Main HUB node, EGFR (degree = 123). In red, HUBs found in the whole net; in yellow, a bottleneck node.

**Figure 9 biomolecules-14-00541-f009:**
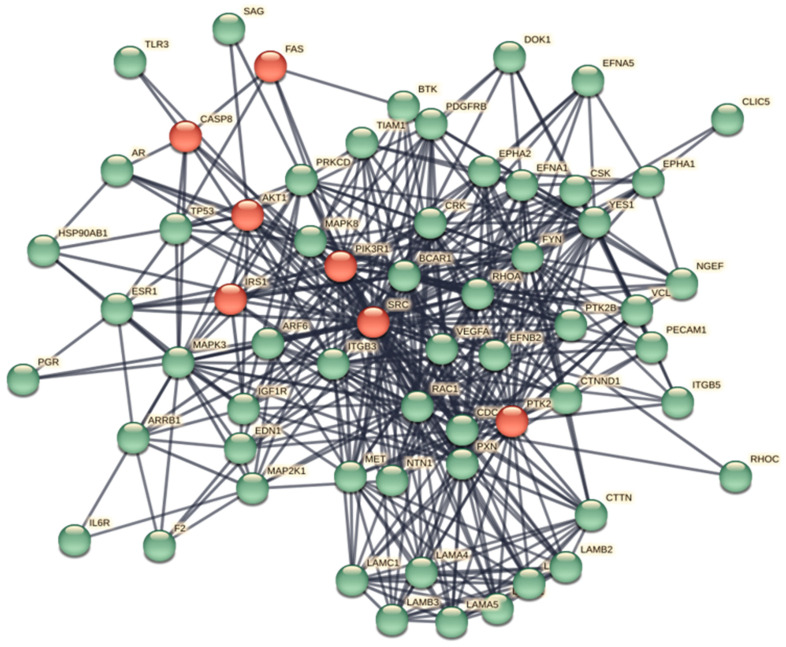
Cluster No. 9—62 nodes, 437 edges, *p*-value < 1.0× 10^−16^. Average node degree 14.097, avg. local clustering coefficient 0.682 (expected number of edges in a similar random network, 83), network diameter 3, network radius 2, characteristic path length 1.798, network density 0.231. Main HUB node, SRC (degree = 56). In red are some of the principal nodes of this cluster.

**Figure 10 biomolecules-14-00541-f010:**
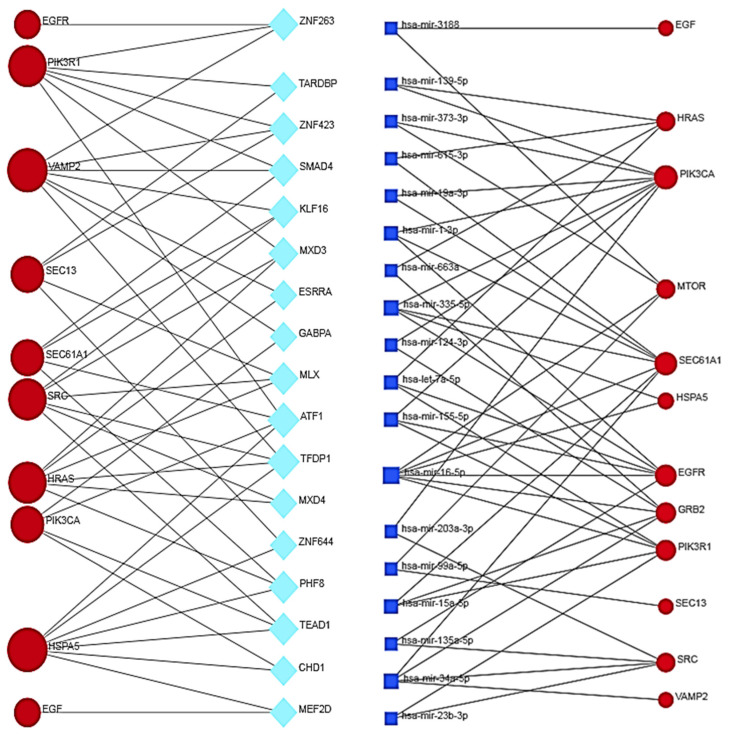
Transcriptional network (**left**) and post-transcriptional network (**right**) of interactions between genes (hubs and bottlenecks) and TFs and miRNAs. Red circles, genes; azure diamonds, TFs; blue rectangles, miRNAs. The rank of nodes in the networks is high as they undergo filtering based on degree and betweenness values. This is only a schematic view of the most significant molecules and their targets, where the size of the node is proportional to its rank.

**Figure 11 biomolecules-14-00541-f011:**
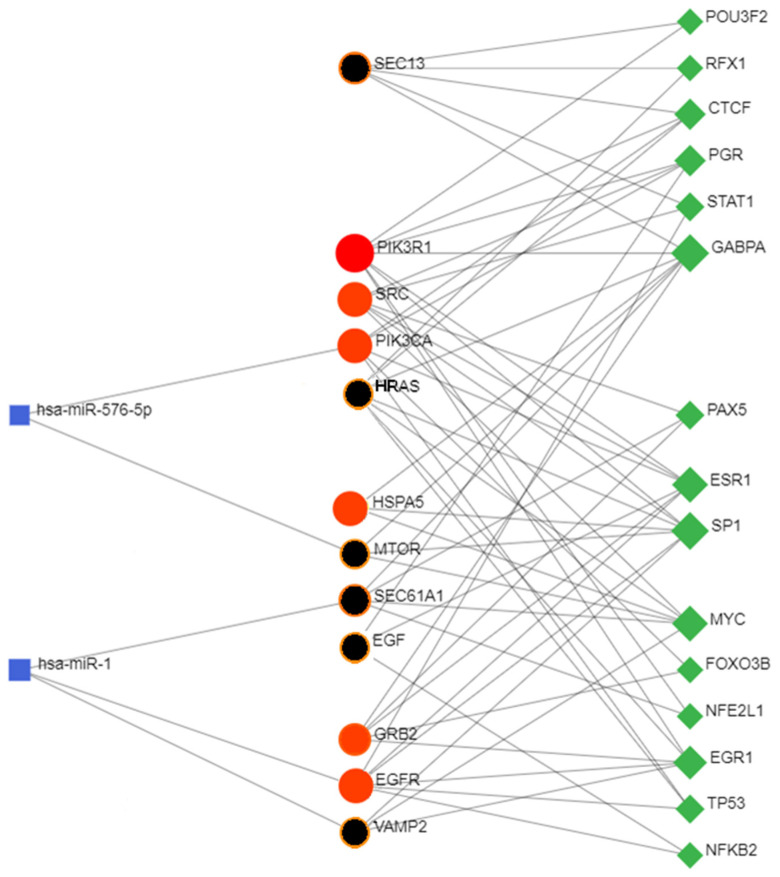
Integrated gene regulatory network associated with dysregulated bottleneck and hub genes. Nodes: red orange circles, hubs; black circles, bottlenecks; green diamonds, TFs; blue rectangles, miRNAs. The figure shows the distribution of the potential gene–TF interactions (center and right side) and gene–miRNA interactions (center and left side). This is only a schematic view of the most significant molecules and their targets. We filtered the interacting network of miRNAs and TFs with betweenness centrality  ≥  100 and 45. Appendix A displays the log–log graph, which confirms a scale-free distribution and shows some topological parameters.

**Figure 12 biomolecules-14-00541-f012:**
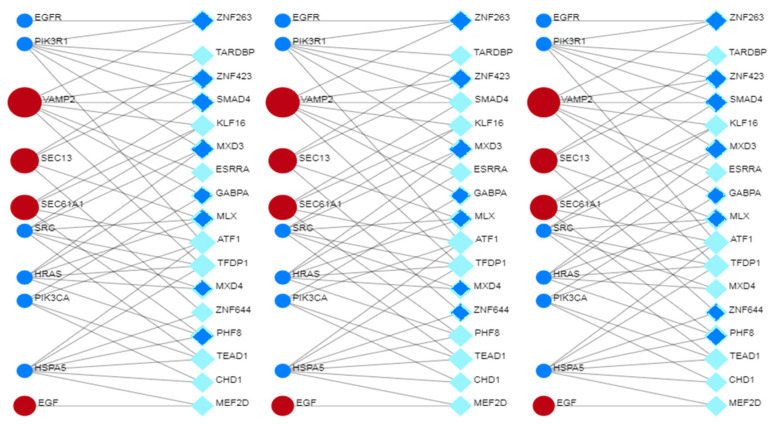
Comparison of three transcriptional networks related to negative metabolic controls because of ORF7b interference. GO analysis (genes in blue, bottlenecks in red). (**Left side**)—Negative regulation of transcription, DNA_dependent (*p* < 1.56 × 10^−4^) (TFs: ZNF263, ZNF423, SMAD4, MXD3, GABPA, MLX, MXD4, PHF8) (**Middle**)—Negative regulation of apoptotic process (*p* < 7.58 × 10^−4^) (TFs: ZNF263, ZNF423, MXD3, GABPA, MLX, MXD4, ZNF644) (**Right side**)—Negative regulation of programmed cell death (*p* < 8.86 × 10^−4^) (TFs: ZNF263, ZNF423, SMAD4, MXD3, GABPA, ZNF644, PHF8).

**Figure 13 biomolecules-14-00541-f013:**
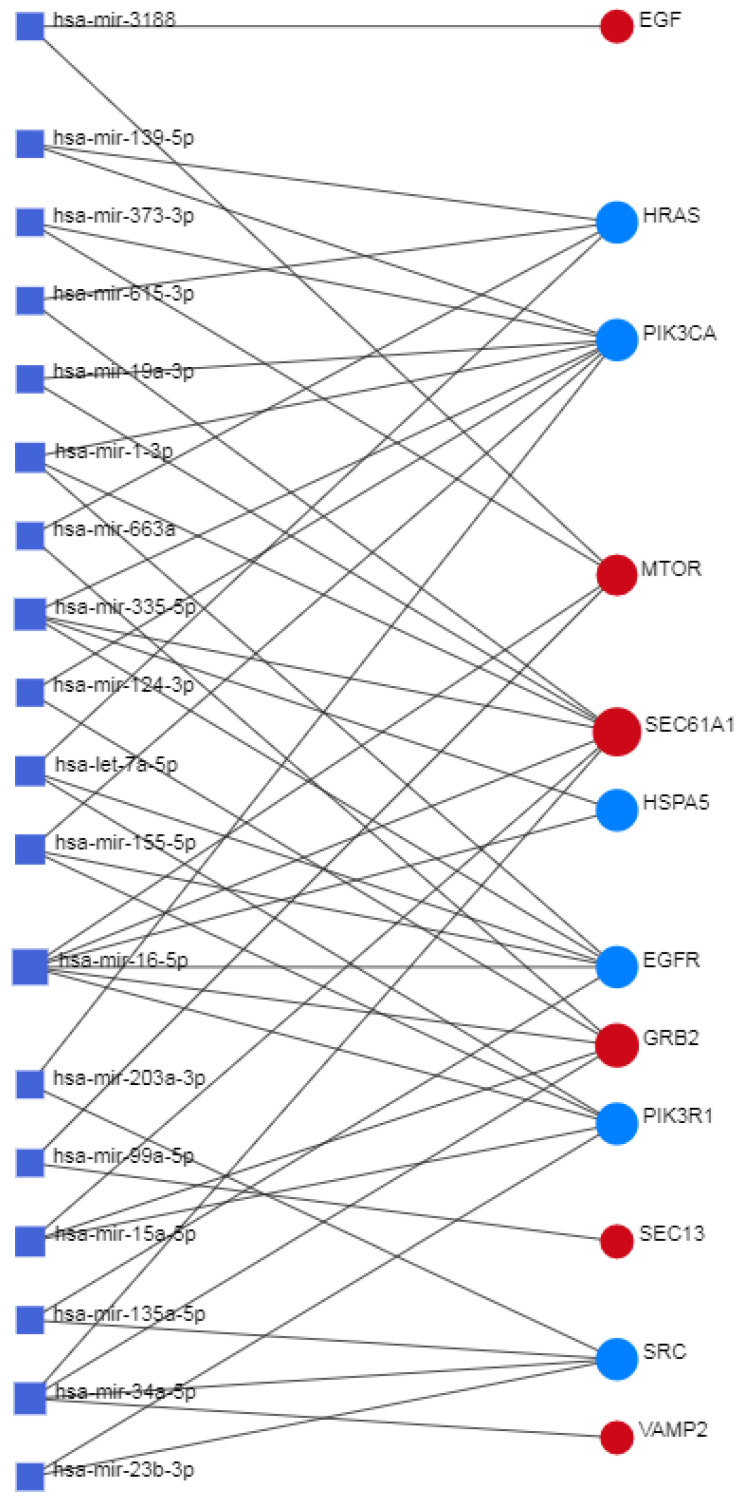
Post-transcriptional networks related to negative metabolic controls because of ORF7b interference. GO analysis for the negative regulation of programmed cell death (*p* < 8.28 × 10^−6^) (EGFR, HRAS, PIK3R1, HSPA5, PIK3CA, SRC).

**Figure 14 biomolecules-14-00541-f014:**
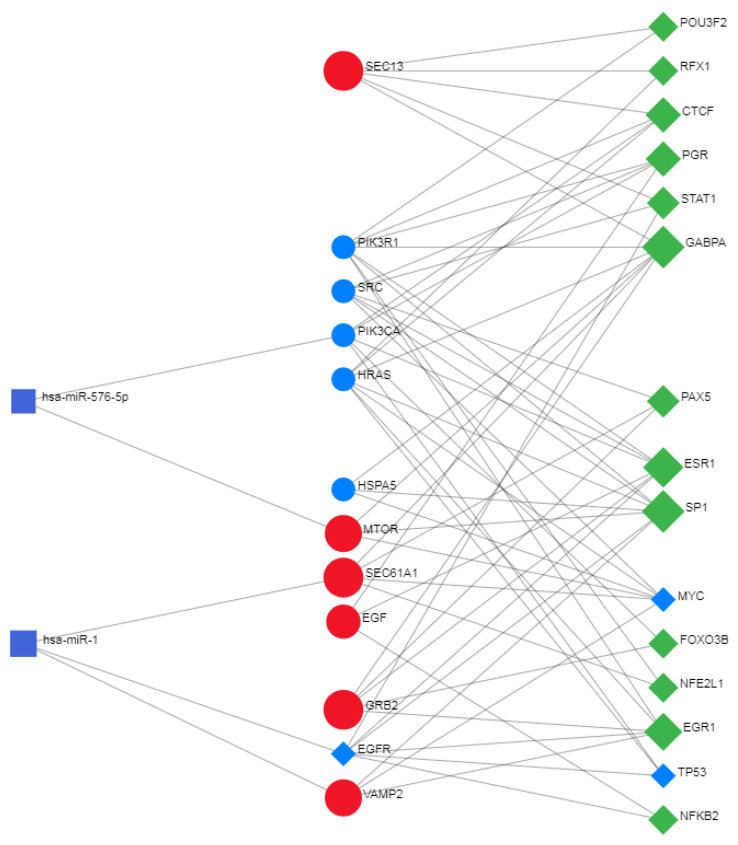
Co-regulated network related to negative metabolic controls because of ORF7b interference. GO analysis for the negative regulation of programmed cell death (*p* < 1.82 × 10^−5^). (TFs: TP53, MYC; Genes: SRC, EGFR, HRAS, PIK3R1, HSPA5, PIK3CA; miRNA: has-miR-1 and has-miR-576-5p).

**Table 2 biomolecules-14-00541-t002:** Functional activities and structural characteristics highlighted by STRING for the interactome of Figure 2.

Action	Enriched Terms
Biological process (Gene Ontology):	1690 GO terms
Molecular function (Gene Ontology):	166 GO terms
Cellular component (Gene Ontology):	267 GO terms
**Reference publications (PubMed):**	**>10,000 publications**
Local network cluster (STRING):	137 clusters
KEGG pathways:	195 pathways
Reactome pathways:	494 pathways
WikiPathways:	259 pathways
Disease–gene associations (DISEASES):	112 diseases
Tissue expression (TISSUES):	186 tissues
Subcellular localization (compartments):	249 compartments significantly
Human phenotype (Monarch):	1002 phenotypes
Annotated keywords (UniProt):	99 keywords
Protein domains (Pfam):	63 domains
Protein domains and features (InterPro):	118 domains
Protein domains (SMART):	20 domains
All enriched terms (without PubMed):	5057 enriched terms in 15 categories

**Table 3 biomolecules-14-00541-t003:** Biological Functions.

GO Term ID	Term Description	Number of Involved Proteins	*p*-Value
GO:0051179	Localization	378	2.01 × 10^−77^
GO:0006810	Transport	320	3.04 × 10^−67^
GO:0007169	Transmembrane receptor protein tyrosine kinase signaling pathway	124	1.23 × 10^−66^
GO:0051234	Establishment of localization	322	7.72 × 10^−66^
GO:0015833	Peptide transport	187	1.09 × 10^−62^
GO:0051649	Establishment of localization in cell	230	3.37 × 10^−62^
GO:0051641	Cellular localization	254	1.29 × 10^−60^
GO:0015031	Protein transport	181	7.86 × 10^−60^
GO:0007167	Enzyme-linked receptor protein signaling pathway	131	7.95 × 10^−59^
GO:0008104	Protein localization	213	1.46 × 10^−58^
GO:0045184	Establishment of protein localization	183	2.85 × 10^−58^
GO:0016192	Vesicle-mediated transport	189	1.18 × 10^−53^
GO:0032879	Regulation of localization	229	2.95 × 10^−51^
GO:0009987	Cellular process	546	4.49 × 10^−51^
GO:0046907	Intracellular transport	168	1.19 × 10^−49^

**Table 4 biomolecules-14-00541-t004:** Cellular localization of biological functions.

GO Term ID	Compartment	Number of Involved Proteins	*p*-Value
GOCC:0016020	Membrane	399	2.58 × 10^−92^
GOCC:0012505	Endomembrane system	302	1.36 × 10^−91^
GOCC:0031090	Organelle membrane	243	2.07 × 10^−77^
GOCC:0098796	Membrane protein complex	189	7.35 × 10^−74^
GOCC:0005737	Cytoplasm	437	1.41 × 10^−73^
GOCC:0031982	Vesicle	213	5.13e × 10^−62^
GOCC:0098588	Bounding membrane of organelle	174	3.17 × 10^−58^
GOCC:0005783	Endoplasmic reticulum	133	6.29 × 10^−55^
GOCC:0098805	Whole membrane	156	1.76 × 10^−53^
GOCC:0110165	Cellular anatomical entity	531	2.59 × 10^−51^
GOCC:0005789	Endoplasmic reticulum membrane	105	4.93 × 10^−51^
GOCC:0042175	Nuclear outer membrane–ER membrane network	106	1.79 × 10^−50^
GOCC:0031410	Cytoplasmic vesicle	177	1.29 × 10^−49^
GOCC:0032991	Protein-containing complex	306	4.76 × 10^−44^
GOCC:0043226	Organelle	437	1.20 × 10^−41^
GOCC:0043227	Membrane-bounded organelle	406	5.80 × 10^−41^
GOCC:0005622	Intracellular	462	8.33 × 10^−38^
GOCC:0043229	Intracellular organelle	407	4.82 × 10^−34^
GOCC:0005829	Cytosol	201	2.18 × 10^−32^
GOCC:0005886	Plasma membrane	220	3.75 × 10^−30^
GOCC:0031201	SNARE complex	34	3.79 × 10^−30^
GOCC:0043231	Intracellular membrane-bounded organelle	349	6.22 × 10^−30^

**Table 5 biomolecules-14-00541-t005:** Reactome.

Term ID	Molecular Mechanism	Number of Involved Proteins	*p*-Value
HSA-9006934	Signaling by receptor tyrosine kinases	140	4.44 × 10^−84^
HSA-1643685	Disease	189	2.66 × 10^−63^
HSA-422475	Axon guidance	101	6.79 × 10^−45^
HSA-9675108	Nervous system development	103	6.79 × 10^−45^
HSA-168256	Immune system	176	6.06 × 10^−41^
HSA-5663205	Infectious disease	115	6.30 × 10^−41^
HSA-162582	Signal transduction	204	2.84 × 10^−37^
HSA-5653656	Vesicle-mediated transport	95	2.70 × 10^−34^
HSA-199991	Membrane trafficking	92	5.34 × 10^−34^
HSA-392499	Metabolism of proteins	163	3.25 × 10^−33^
HSA-109582	Hemostasis	89	1.02 × 10^−32^
HSA-1799339	SRP-dependent cotranslational protein targeting to membrane	45	2.19 × 10^−31^
HSA-168249	Innate immune system	111	1.33 × 10^−30^
HSA-1227986	Signaling by ERBB2	35	2.43 × 10^−30^
HSA-74752	Signaling by insulin receptor	38	5.77 × 10^−29^
HSA-177929	Signaling by EGFR	33	5.40 × 10^−28^
HSA-4420097	VEGFA-VEGFR2 pathway	39	5.35 × 10^−27^
HSA-202733	Cell surface interactions at the vascular wall	42	3.19 × 10^−25^
HSA-76002	Platelet activation, signaling, and aggregation	52	4.89 × 10^−24^
HSA-6811558	PI5P, PP2A, and IER3 regulate PI3K/AKT signaling	37	5.16 × 10^−24^
HSA-5683057	MAPK family signaling cascades	52	1.19 × 10^−20^
HSA-5684996	MAPK1/MAPK3 signaling	49	1.37 × 10^−20^
HSA-77387	Insulin receptor recycling	21	1.07 × 10^−19^
HSA-192823	Viral mRNA translation	27	1.54 × 10^−16^
HSA-1500931	Cell–cell communication	30	1.91 × 10^−15^

**Table 6 biomolecules-14-00541-t006:** Human tissues involved with ORF7b.

TERM ID	Human Tissues Involved with ORF7b	Number of Involved Proteins	*p*-Value
BTO:0000345	Digestive gland	233	4.73 × 10^−56^
BTO:0001491	Viscus	322	1.59 × 10^−54^
BTO:0001489	Whole body	504	2.18 × 10^−45^
BTO:0000522	Gland	356	2.76 × 10^−45^
BTO:0000759	Liver	178	2.17 × 10^−44^
BTO:0001488	Endocrine gland	323	1.15 × 10^−37^
BTO:0003091	Urogenital system	341	3.03 × 10^−36^
BTO:0000227	Central nervous system	303	1.41 × 10^−35^
BTO:0001484	Nervous system	307	3.23 × 10^−35^
BTO:0000449	Fetus	125	1.68 × 10^−32^
BTO:0001078	Placenta	119	1.40 × 10^−30^
BTO:0000081	Reproductive system	308	5.01 × 10^−30^
BTO:0003099	Internal female genital organ	183	5.01 × 10^−30^
BTO:0000174	Embryonic structure	159	7.75 × 10^−28^
BTO:0000203	Respiratory system	127	9.81 × 10^−28^
BTO:0000083	Female reproductive system	292	1.71 × 10^−27^
BTO:0000089	Blood	136	1.39 × 10^−26^
BTO:0000570	Hematopoietic system	172	6.39 × 10^−26^
BTO:0000763	Lung	105	3.59 × 10^−23^
BTO:0000988	Pancreas	72	6.06 × 10^−23^
BTO:0000431	Excretory gland	106	8.44 × 10^−21^
BTO:0003092	Urinary system	97	3.43 × 10^−19^
BTO:0001244	Urinary tract	97	4.17 × 10^−19^
BTO:0000671	Kidney	86	5.49 × 10^−19^
BTO:0001129	Prostate gland	58	8.08 × 10^−19^
BTO:0000132	Blood platelet	50	2.81 × 10^−18^
BTO:0000511	Gastrointestinal tract	116	4.50 × 10^−17^
BTO:0000131	Blood plasma	51	1.46 × 10^−16^
BTO:0000574	Hematopoietic cell	77	1.21 × 10^−14^
BTO:0000082	Male reproductive system	148	3.21 × 10^−14^
BTO:0000751	Leukocyte	72	3.31 × 10^−14^
BTO:0000080	Male reproductive gland	138	6.39 × 10^−13^
BTO:0000254	Female reproductive gland	145	7.61 × 10^−12^
BTO:0005810	Immune system	96	3.68 × 10^−11^
BTO:0003096	Internal male genital organ	122	4.92 × 10^−11^
BTO:0000088	Cardiovascular system	70	4.40 × 10^−10^
BTO:0000421	Connective tissue	63	1.19 × 10^−09^
BTO:0000439	Eye	59	1.33 × 10^−09^
BTO:0000706	Large intestine	54	1.57 × 10^−09^
BTO:0000202	Sense organ	69	1.98 × 10^−09^
BTO:0000855	Lymph	25	4.56 × 10^−09^
BTO:0001085	Vascular system	38	9.90 × 10^−09^
BTO:0001424	Uterus	67	1.11 × 10^−08^
BTO:0000269	Colon	46	3.05 × 10^−08^
BTO:0001363	Testis	85	2.48 × 10^−05^

**Table 7 biomolecules-14-00541-t007:** Most significant KEGG pathways in the human interactome induced by ORF7b.

Pathway	Description	Number of Involved Proteins	*p*-Value
hsa04012	ErbB signaling pathway	50	3.02 × 10^−41^
hsa04510	Focal adhesion	64	2.27 × 10^−40^
hsa01521	EGFR tyrosine kinase inhibitor resistance	46	4.29 × 10^−38^
hsa04151	PI3K-Akt signaling pathway	74	1.16 × 10^−36^
hsa04141	Protein processing in ER	55	1.44 × 10^−35^
hsa04015	Rap1 signaling pathway	51	3.72 × 10^−28^
hsa04014	Ras signaling pathway	52	3.76 × 10^−27^
hsa05206	MicroRNAs in cancer	45	1.48 × 10^−26^
hsa04935	Growth hormone synthesis, secretion action	40	3.80 × 10^−26^
hsa04130	SNARE interactions in vesicular transport	27	1.38 × 10^−25^
hsa04062	Chemokine signaling pathway	45	2.35 × 10^−24^
hsa04145	Phagosome	40	9.33 × 10^−24^
hsa04360	Axon guidance	43	2.15 × 10^−23^
hsa04072	Phospholipase D signaling pathway	39	2.11 × 10^−22^
hsa04917	Prolactin signaling pathway	30	2.84 × 10^−22^
hsa04150	mTOR signaling pathway	39	4.38 × 10^−22^
hsa04810	Regulation of actin cytoskeleton	42	3.07 × 10^−20^
hsa01522	Endocrine resistance	31	4.10 × 10^−20^
hsa04915	Estrogen signaling pathway	35	4.10 × 10^−20^
hsa04722	Neurotrophin signaling pathway	33	4.29 × 10^−20^
hsa04919	Thyroid hormone signaling pathway	32	9.72 × 10^−19^
hsa04664	Fc epsilon RI signaling pathway	26	1.16 × 10^−18^
hsa04010	MAPK signaling pathway	45	5.16 × 10^−18^
hsa04721	Synaptic vesicle cycle	26	1.05 × 10^−17^
hsa04660	T cell receptor signaling pathway	29	1.05 × 10^−17^
hsa04662	B cell receptor signaling pathway	26	2.90 × 10^−17^
hsa04650	Natural killer cell-mediated cytotoxicity	30	7.46 × 10^−17^

**Table 8 biomolecules-14-00541-t008:** Dysregulated processes related to the cytoskeleton.

GO Biological Process	Description	P	Strength	fdr	Color
GO:2001237	Negative regulation of extrinsic apoptotic signaling pathway	8.05	1.18	6.60 × 10^−6^	
GO:0051129	Negative regulation of cellular component organization	7.23	0.76	3.79 × 10^−9^	
GO:2000811	Negative regulation of anoikis	7.00	1.58	2.70 × 10^−4^	
GO:0043069	Negative regulation of programmed cell death	6.66	0.70	3.24 × 10^−9^	
GO:0060548	Negative regulation of cell death	6.50	0.67	4.93 × 10^−9^	
GO:0048519	Negative regulation of biological process	5.62	0.39	2.67 × 10^−14^	
GO:0043066	Negative regulation of apoptotic process	5.54	0.69	1.09 × 10^−8^	
GO:0023057	Negative regulation of signaling	5.13	0.58	7.04 × 10^−8^	
GO:0010648	Negative regulation of cell communication	5.11	0.58	6.42 × 10^−8^	
GO:0031333	Negative regulation of protein-containing complex assembly	4.56	0.95	6.40 × 10^−4^	
GO:0010507	Negative regulation of autophagy	4.42	1.10	4.50 × 10^−4^	
GO:0032369	Negative regulation of lipid transport	4.23	1.39	1.10 × 10^−3^	
GO:2001234	Negative regulation of apoptotic signaling pathway	3.74	0.85	2.50 × 10^−4^	
GO:1902904	Negative regulation of supramolecular fiber organization	2.80	0.89	1.40 × 10^−3^	
GO:0051494	Negative regulation of cytoskeleton organization	2.77	0.90	1.20 × 10^−3^	
GO:0007162	Negative regulation of cell adhesion	2.19	0.75	1.20 × 10^−3^	

**Table 9 biomolecules-14-00541-t009:** Involvement of HUBs and bottlenecks in the control of biological processes (GO).

HUB Protein	Number of GO Processes	Bottleneck Protein	Number of GO Processes
EGFR	408	EGFR	408
PIK3R1	328	HSPA5	234
EGF	646	MTOR	413
HRAS	245	SEC13	83
GRB2	233	SEC61A1	63
SRC	508	SRC	508
PIK3CA	271	VAMP2	143

Note: EGFR and SRC are on both lists because of their dual activity. From the genes paired to each term, STRING extracted the biological processes for each individual protein under the biological processes (GO) section.

**Table 10 biomolecules-14-00541-t010:** Multiple interactions of EGFR, SRC, and PIK3R1 with viral proteins.

Viral Protein	Human Target	Viral Protein Features **
nsp4 *	EGFR	Involved in the assembly of virally induced cytoplasmic double-membrane vesicles necessary for viral replication
M *	EGFR	Component of the viral envelope
ORF3a *	EGFR	Homotetrameric potassium-sensitive ion channels (viroporin) that may modulate virus release
ORF7b *	EGFR	This paper
S	EGFR	Spike or surface glycoprotein.
nsp4 *	SRC	See above
nsp5 *	SRC	A cysteine protease essential for the viral life cycle
nsp6 *	SRC	Plays a role in the initial induction of auto-phagosomes from host reticulum endoplasmic
nsp13 *	SRC	Multi-functional helicase with a zinc-binding domain in N-terminus
nsp14 *	SRC	3′-5′ deoxyribonuclease
E *	SRC	Plays a central role in virus morphogenesis and assembly
M *	SRC	See above
ORF3a *	SRC	See above
ORF3b	SRC	Could be involved in immune evasion as an interferon agonist ***
ORF6 *	SRC	Could be a determinant of virus virulence
ORF7a *	SRC	A non-structural protein, which is dispensable for virus replication in cell culture
ORF7b *	SRC	See above
ORF8	SRC	A viral cytokine regulating immune responses
S	SRC	See above
M *	PIK3R1	See above
ORF7b *	PIK3R1	See above
ORF3b	PIK3R2	See above
M *	PIK3R3	See above
S	PIK3R3	See above
N *	ORF7b	Responsible for wrapping viral RNA into a symmetric helical structure

Notes: (*) With a few exceptions (ORF3b, ORF8, and S), all the remaining viral proteins, although compact, have intrinsically disordered regions (IDRs), often in the tails, which make interactions with numerous partners possible. This could be the structural cause of their multiple actions. (**) Viral protein information from National Center for Biotechnology Information, the National Library of Medicine, USA. (***) [100].

**Table 11 biomolecules-14-00541-t011:** Commonly altered pathways.

Function	Strength *	*p*-Value	P	Human Proteins Involved in the Process
Negative regulation of the ERBB signaling pathway	1.22	1.38 × 10^−18^	22.13	HBEGF, EREG, PTPN12, TSG101, CBL, CBLB, **EGF**, ERBB2, CBLC, **EGFR**, TGFA, SOCS5, PTPN2, HGS, EPS15, ERRFI1, SNX5, SH3GL2, **GRB2**, BTC, AREG, SH3KBP1, CDC42, EPN1, EPGN
Negative regulation of the EGFR signaling pathway	1.23	3.24 × 10^−17^	21.53	HBEGF, EREG, TSG101, CBL, CBLB, **EGF**, CBLC, **EGFR**, TGFA, SOCS5, PTPN2, HGS, EPS15, ERRFI1, SNX5, SH3GL2, **GRB2**, BTC, AREG, SH3KBP1, CDC42, EPN1, EPGN
Negative regulation of anoikis	1.14	5.72 × 10^−5^	6.56	**PIK3CA**, **ITGA5**, **BCL2L1**, **CAV1**, **PTK2**, **SRC**, **ITGB1**
Negative regulation of the extrinsic apoptotic signaling pathway	0.76	9.26 × 10^−7^	6.05	GCLC, LGALS3, BCL2L1, IGF1, CTNNA1, UNC5B, FYN, **FAS, CASP8**, LMNA, GCLM, **SRC**, AR, CTTN, NRG1, **ITGA6**, **AKT1**
Negative regulation of protein tyrosine kinase activity	0.99	9.13 × 10^−5^	5.90	TSG101, CBL, CBLB, CBLC, SOCS5, PTPN2, **CAV1**, ERRFI1
Negative regulation of epidermal growth factor-activated receptor activity	1.18	1.7 × 10^−4^	4.99	TSG101, CBL, CBLB, CBLC, SOCS5, ERRFI1
Negative regulation of interleukin-6 production	0.81	5.0 × 10^−5^	4.61	CSK, SOCS5, GAS6, TLR9, VIMP, PTPN6, ARRB1, ENSP00000417517
Negative regulation of peptidyl-tyrosine phosphorylation	0.88	1.50 × 10^−5^	4.55	TSG101, CBL, CBLB, CBLC, SPINK1, SOCS5, PTPN2, CAV1, ERRFI1, PRKCD, PTPN6
Negative regulation of PERK-mediated unfolded protein response	1.33	9.2 × 10^−3^	4.12	NCK2, PTPN1, NCK1
Negative regulation of endoplasmic reticulum unfolded protein response	1.04	8.9 × 10^−3^	4.11	NCK2, **HSPA5**, PTPN1, NCK1
Negative regulation of blood–brain barrier permeability	1.55	3.13 × 10^−2^	3.88	SH3GL2, VEGFA
Negative regulation of response to oxidative stress	0.81	4.1 × 10^−4^	3.74	SLC7A11, MET, GGT7, CTNNB1, FYN, NFE2L2, **INS**, HIF1A, **AKT1**
Negative regulation of protein tyrosine phosphatase activity	1.39	4.67 × 10^−2^	3.71	LGALS3, GNAI2
Negative regulation of mesenchymal to epithelial transition	1.38	4.77 × 10^−2^	3.69	CTNNB1, STAT1
Negative regulation of blood coagulation	0.83	2.8 × 10^−4^	3.69	PROC, PDGFRA, F2, PLAUR, PLAU, EDN1, CD9, PROS1, PRKCD
Negative regulation of primary miRNA processing	1.38	4.67 × 10^−2^	3.68	STAT3, IL6
Negative regulation of lipid transport	0.79	1.74 × 10^−2^	1.77	**EGF**, PTPN11, SREBF2, AKT1, ITGB3

Note: (*) For comparisons, a strength of 1.39 = 24.5 times enrichment, and 0.76 = 5.8 times. Black bold font indicates the proteins highlighted in the text as being HUB nodes or “bottlenecks” or as being involved in other important signaling pathways.

**Table 13 biomolecules-14-00541-t013:** Operational cellular compartments of cluster No. 9 proteins.

Compartment	Proteins *	Protein Number
Extracellular	EDN1, **F2**, **FAS**, HSP90AB1, LAMA5, LAMC1, **MET**, NTN1, **VEGFA**	9
Cytoskeleton	CDC42, **CTNNB1**, CTTN, LMNA, MAPK3, **PTK2**, PXN, YES1	8
Plasma membrane	**AKT1**, ARF6, **CASP8**, **CAV1**, **CDC42**, CDH1, CDH2, **CTNNB1**, CTNND1, EFNA5, EFNB2, EPHA1, EPHA2, ESR1, **FAS**, HRAS, IGF1R, **ITGB3**, **MET**, NEDD4, PDGFRB, PECAM1, **PRKCD**, **PTK2**, PTK2B, PXN, RAC1, RHOA, **SRC**, TIAM1, TJP1, YES1	32
Cytosol	**AKT1**, ARF6, **CASP8**, **CTNNB1**, MAPK3, **PRKCD**, **PTK2**, RHOA, **SRC**, YES1,	10
Mitochondrion	GJA1, HSP90AA1, MAP2K1, MAPK3, **SRC**	5
Golgi	**CBL**, CDH1, ESR1, HRAS, MAP2K1, MAPK3, NEDD4, RAC1, YES1	9
Endoplasmic Reticulum	**PRKCD**, MAP2K1	2
Peroxisome	No level 5 protein	-
Endosome	ARF6, **CAV1**, CDH1, MAP2K1, MAPK3, **PRKCD**, RAC1, **SRC**	8
Lysosome	PDGFRB, PRKCD, **SRC**	3
Nucleus	**AKT1**, **AR**, **ARRB1**, **CTNNB1**, ESR1, GJA1, HRAS, HSP90AB1, **ITGB3**, **LMNA**, MAP2K1, MAPK8, NEDD4, PGR, **PRKCD**, **PTK2**, PTK2B, RAC1, **STAT3**	19

Note: (*) Only proteins with a statistical significance value of 5, according to Cytoscape analysis. These values range from 0 to 5. We calculated protein node compartmentalization and values in Cytoscape using the STRING app. In bold font are the proteins in common with Table 9.

## Data Availability

Data are contained within the article and Appendix A.

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
