# Peer review of "A Tiny Viral Protein, SARS-CoV-2-ORF7b: Functional Molecular Mechanisms"

_biomolecules, 2024, doi:10.3390/biom14050541_

Round 1

Reviewer 1 Report

Comments and Suggestions for Authors

Reviewer report

The manuscript presented by Gelsomina Mansueto, Giovanna Fusco and Giovanni Colonna addresses an interesting topic regarding the protein-protein interactions that ORF7b of SARS-CoV-2 may have. This is a totally in silico study, using different bioinformatics processes that use curated databases generated by third parties as data. However, the manuscript has several problems that will need to be addressed:

1. The manuscript is excessively long, written in unclear language and with unnecessary details in different sections.

2. Due to the length, the primary objectives of the work seem to be diluted, so no substantial conclusions are reached. Of course, the discussion does not contribute much to clear objectives either, due to this lack of clarity.

3. Although methodologically it seems to be supported, there is a great lack of clarity in various processes, especially in the statistical part. I understand that each computer process has a particular statistical process, but the authors must declare each process in its corresponding section.

Below, I detail some of these observations regarding the manuscript and the corresponding line number:

Materials and methods

Lines 83-87: The authors should clarify what implications the statistical analysis of protein enrichment has, since it is not clear whether the fact that the "protein enrichment is to some extent based on prior knowledge, and the statistical enrichment of the annotated features may not be an intrinsic property of the input" is a statistical disadvantage for the job workflow.

Lines 113-116

The authors must find a way to synthesize the mathematical analysis section. While I understand the purpose is to clarify the methodology, I suggest going straight to a supplemental file and adhering to properly written equations.

Lines 125-126: That is a result, it is not part of the method.

Lines 151-161: Place this information in a table.

Lines 186: Why not show these results?

196: This seems a serious problem to me since it compromises the integrity of all the results. Authors should clearly demonstrate why it would not compromise all results.

204-206: The authors should improve the language.

222-241: It is not understandable. The authors should rewrite the paraghraph using appropriate mathematical language.

245-257: I don't understand who these results are from, the database, or the authors who have gathered this information?

Figure 1: The figure is not understandable, it does not have an adequate resolution.

Lines 378-380: What tests do the authors propose? Since it is intended to be an in silico study, lines consistent with its results should at least be suggested.

In summary, the manuscript presented has serious problems of length, comprehensibility, language (mathematical, tables, figures), lack of methodological consistency, results, and consequently, the discussion presented has little or no scientific value, in addition to being excessively speculative and imprecise. The authors must rewrite the entire manuscript, evaluate the relevance of the methodology used and present a definitively improved work.

Author Response

Reviewer 1 report

The manuscript presented by Gelsomina Mansueto, Giovanna Fusco and Giovanni Colonna addresses an interesting topic regarding the protein-protein interactions that ORF7b of SARS-CoV-2 may have. This is a totally in silico study, using different bioinformatics processes that use curated databases generated by third parties as data. However, the manuscript has several problems that will need to be addressed:

Response: Dear reviewer 1, thank you for your comments. We respond to your comments and highlight new additions in red in the text.

  1. The manuscript is excessively long, written in unclear language and with unnecessary details in different sections.

Response: We have revised the English language and reduced the length. You will find the changes in red.

  1. Due to the length, the primary objectives of the work seem to be diluted, so no substantial conclusions are reached. Of course, the discussion does not contribute much to clear objectives either, due to this lack of clarity.

Response: we have reduced the length and better defined the objectives in the conclusion. You will find the changes in red. Our study has produced a model of the main deep metabolic pathways, highlighting the most important components and which are the viral proteins that attack them.

  1. Although methodologically it seems to be supported, there is a great lack of clarity in various processes, especially in the statistical part. I understand that each computer process has a particular statistical process, but the authors must declare each process in its corresponding section.

Response: In the manuscript we report the computational analyzes on experimental data selected among the most significant present in BioGRID. BioGRID offers a significance score for each reported interaction. All the analyzes reported by us were done exclusively on experimental data using mathematical algorithms that always report the statistical evaluation. The network shown has a p-value <10e-16. All the GO functional analyzes derive from it, of which we have reported the most significant ones with statistical values always between <1e-92 and <1e-15 (except for table VI). We don't see what else we can add. In Section II of the Supplements, we explain on what basis the robustness of our study is based.

Below, I detail some of these observations regarding the manuscript and the corresponding line number:

 Materials and methods

Lines 83-87: The authors should clarify what implications the statistical analysis of protein enrichment has, since it is not clear whether the fact that the "protein enrichment is to some extent based on prior knowledge, and the statistical enrichment of the annotated features may not be an intrinsic property of the input" is a statistical disadvantage for the job workflow.

Response:

Computer algorithms, while being powerful tools for data analysis and information processing, depend on the quality of the data and information they receive. If the data is polluted or distorted, the conclusions and answers provided may be equally unreliable. Therefore, it is essential to take measures to mitigate the risk of polluting scientific knowledge, such as accurate collection and curation of data, verification and replication of results, transparency and open sharing of methods and data used in analyses. We need to rigorously review and verify the data and ensure transparency and reproducibility. In their absence, algorithms are of little use.

To this end, the "protein enrichment is to some extent based on prior knowledge, and the statistical enrichment of the annotated features may not be an intrinsic property of the input" means that STRING to quantitatively analyze the relationships existing between the nodes deriving from the enrichment, creates a specific knowledge base that collects from the literature all the data and information necessary for the calculation of the network and its statistics. Prior knowledge means derived from knowledge already deposited in the literature. Our network models all have p-values far <10e-16 and the reported functional relationships (GO) are all characterized by extremely low statistical values.

To obtain all this, STRING analyzes the entire PubMed using AI and selecting all the existing scientific articles on the interactions that are the subject of the interactome under study. The network model we presented in Figure 2 was calculated based on information and data from over 10,000 articles examined by STRING (see Table II, under the heading “Reference Publications”). The entire list of these articles can be downloaded from STRING and is provided with a p-value for each individual article. “Annotated features” are the hyperparameters (metadata) containing experimental and numerical data that database curators around the world should add to the metadata to complete the information. Unfortunately, they don't always do this. In conclusion, we can say that we have worked on a robust set of data and information.

Lines 113-116  [“Integrating network topological quantification with other numerical node attributes can cause relevant node identification and functional classification, as well as the topological location of proteins in their specific cellular compartments.”]

The authors must find a way to synthesize the mathematical analysis section. While I understand the purpose is to clarify the methodology, I suggest going straight to a supplemental file and adhering to properly written equations.

Response: Table 1 and figure 5 show all the topological parameters and node distributions calculated through Cytoscape.

Lines 125-126: That is a result, it is not part of the method.

Response: We thank the referee for this observation. The final part of paragraph 2.3 has been transferred to 3.3 in the comment of figure 3.

 Lines 151-161: Place this information in a table.

and

Lines 186: Why not show these results?

“existing relationships between genes, proteins, miRNAs, and human transcription factors, creating a co-regulatory network that is very useful for understanding the mutual relationships between these biological actors.”

Response: Paragraph 2.6 NetworkAnalyst, reports the description of the methods used to analyze the relationships existing with the miRNAs used. All results are shown in figures 10, 11,12, 13, and 14.

196: This seems a serious problem to me since it compromises the integrity of all the results. Authors should clearly demonstrate why it would not compromise all results.

“Upon visually inspecting the graphic outputs of both predictive systems, we quickly identified disordered segments in most of the examined proteins, whether viral or human. These results were not displayed because they required a large space.”

Response: The results calculated for all the proteins of the interactome are shown in a graph in Section II of the Supplements, from which the overall information in % terms on the entire population of the interactome is obtained. As can be seen, many of the proteins have high percentages of intrinsic disorder. Reporting the characteristics of single proteins means reporting numerous graphs that take up space.

204-206: The authors should improve the language.

Response: we have improved the language and reduced the test, as you suggested. The variations are in red.

222-241: It is not understandable. The authors should rewrite the paraghraph using appropriate mathematical language.

Response: This paragraph has been rewritten and shortened to make its meaning clear.

245-257: I don't understand who these results are from, the database, or the authors who have gathered this information?

Response: The paragraph has been rewritten and shortened. The calculations are done by us on STRING data.

Figure 1: The figure is not understandable, it does not have an adequate resolution.

Response: The figure comes from BioGRID which gives us permission to download it. Unfortunately, the definition is that of BioGRID even if we have improved it as much as possible. However, its meaning is not so much to read the name of each node but to show the enormous number of human proteins that physically interact with ORF7b. The related protein list can be easily found in BioGRID.

Lines 378-380: What tests do the authors propose? Since it is intended to be an in silico study, lines consistent with its results should at least be suggested.

“The large diameter, network heterogeneity, and low-density support this view [29]. The diameter also suggests components quite distant from the central module.”

Response: This part has been explained in the text with more details. The mentioned topological parameters, numerically shown in Table I and in some graphs of the manuscript, provide us with a guide on how to continue our investigations. In fact, at the end of paragraph 3.3 we explained what information these data lead us to look for. Computational analysis finds molecular relationships, and the molecules involved, at a metabolic level so deep where no "deterministic" technique can reach. The current limit is the space-time characteristics but an evolution of the technique called "single cell" which allows sampling over time from a single cell without destroying it, will give us information that we don't have today.

In summary, the manuscript presented has serious problems of length, comprehensibility, language (mathematical, tables, figures), lack of methodological consistency, results, and consequently, the discussion presented has little or no scientific value, in addition to being excessively speculative and imprecise. The authors must rewrite the entire manuscript, evaluate the relevance of the methodology used and present a definitively improved work.

Response: With your suggestions we have remodeled the manuscript. Let's hope everything is better now.

Reviewer 2 Report

Comments and Suggestions for Authors

The author has done a great job with the manuscript. However, I do have some suggestions and comments:

1.     The text doesn't explicitly mention potential limitations or challenges associated with the experimental methods used.

2.     While the text extensively discusses the interactome and potential functional implications, it doesn't explicitly mention experimental validations or functional assays conducted to confirm the biological relevance of the observed interactions.

3.     The manuscript provides a snapshot of the interactions and activities without discussing the temporal aspects of these processes. Understanding how these interactions change over time could provide insights into the dynamics of the virus-host interaction.

4.     The readers may benefit if the authors consider and discuss alternative explanations for the observed interactions. Exploring different hypotheses and addressing potential alternative interpretations could strengthen the scientific rigor of the study.

5.     While BioGRID is a valuable resource, it might be beneficial to integrate findings from other independent datasets or experiments to validate and corroborate the observed interactions.

6.     The text could expand on the potential clinical relevance of the findings. How do the observed interactions relate to the clinical manifestations of COVID-19, and are there implications for therapeutic interventions?

7.     While ORF7b is discussed in the context of interactions, the text could delve deeper into the known functions of ORF7b and how the observed interactions align with or contribute to its established roles.

Author Response

Reviewer 2 report

The author has done a great job with the manuscript. However, I do have some suggestions and comments:

Response: Dear reviewer, thank you for your comments. We respond to your concerns remodeling great part of the manuscript (in red). .

  1. The text doesn't explicitly mention potential limitations or challenges associated with the experimental methods used.

Response: In the Supplements and in particular in Section II we explained the robustness and limitations of our investigations. Even in the conclusions we gave explanations on the meaning of our work.

  1. While the text extensively discusses the interactome and potential functional implications, it doesn't explicitly mention experimental validations or functional assays conducted to confirm the biological relevance of the observed interactions.

Response: this is precisely the fulcrum of the work carried out by the authors which, however, is experimental and a harbinger for future functional tests. We selected only highly significant data because the validations and significance were all already present in both BioGRID and STRING and are always reported in the text and supplements. None of the data we report has a p-value that is not highly significant. The network has a p<1.0e-16 and highly significant values are always reported in all tables. The interactions curated on BioGRID are all experimental and meaningful. The same happens for STRING, with the advantage that having 7 channels, it allows you to select the type of data you want to use for the analyses.

  1. The manuscript provides a snapshot of the interactions and activities without discussing the temporal aspects of these processes. Understanding how these interactions change over time could provide insights into the dynamics of the virus-host interaction.

Response: Measurements on the same cellular sample made at different times are not suitable for bringing the time factor into interactomic analyzes because the metabolic context of the cells changes dynamically over time and is always different. The approach we used highlights molecular processes at a deep metabolic level (the degenerate one) that cannot be obtained with other approaches. The macroscopic aspects are properties called emergent, not predictable and connected in a non-linear way with deep processes.

To measure how the interactions between molecules change over time, measurements must be made on a single cell at different times. To be clear, we means measurements always on the same single cell. Current Single cell techniques actually destroy the cell and the metabolism of another single cell will never be the same. Only for a very short time (months) has a sophisticated and brand-new technology been developed that allows multiple sampling over time on a single cell without destroying it.

4 The readers may benefit if the authors consider and discuss alternative explanations for the observed interactions. Exploring different hypotheses and addressing potential alternative interpretations could strengthen the scientific rigor of the study.

Response: We have, where possible, made functional hypotheses that had a certain significance. It is a very delicate topic, because the characteristics of the holistic approach offer great possibilities for dreaming but we must suggest significant and real possibilities (which we have done). Those who must implement it are those who use the methods of biochemistry, biophysics or pathophysiology in their laboratories and then pass the results on to clinicians.

5 While BioGRID is a valuable resource, it might be beneficial to integrate findings from other independent datasets or experiments to validate and corroborate the observed interactions.

Response: Data from other databases are poorly integrable with BioGRID data, both because BioGRID only uses physical interactions, experimentally proven, and because each database has different purposes that are imposed by different curators. Almost no one collects experimental data and validates it. All this is clearly understandable in the manuscript.

6 The text could expand on the potential clinical relevance of the findings. How do the observed interactions relate to the clinical manifestations of COVID-19, and are there implications for therapeutic interventions?

Response: The aim of the work as explained above is to indicate the metabolic effects caused by ORF7b. Some clinical clues were given that had concrete reliability, such as the spread of infected cells with mechanisms that mimic tumor metastasis. We gave precise evaluations in Table VI, on infectible tissues, and in Tables VIII and XI on dysregulated molecular mechanisms. We are studying these mechanisms even more deeply, to pinpoint in which cellular compartments these high-ranking molecules operate exclusively. The first results are overturning many things already known, but this will be the subject of future articles.

7 While ORF7b is discussed in the context of interactions, the text could delve deeper into the known functions of ORF7b and how the observed interactions align with or contribute to its established roles.

Response: the text is already too long and demanding. We showed a deep molecular landscape and its very broad dysregulations. Apparently, all caused by a single protein. We are now reducing the manuscript to better focus the findings. However, we reserve the right to delve deeper into these aspects in a subsequent article because ORF7b shows very complex and varied aspects in its action.

Round 2

Reviewer 1 Report

Comments and Suggestions for Authors

Thank you very much for attend all the comments